



# Spatially-explicit estimates of global wastewater production, collection, treatment and re-use.

Edward R. Jones[1], Michelle T.H. van Vliet[1], Manzoor Qadir[2], Marc F.P Bierkens[1,3]

[1]Department of Physical Geography, Faculty of Geosciences, Utrecht University, Utrecht, The Netherlands.
[2]United Nations University: Institute for Water, Environment and Health (UNU-INWEH), Hamilton, Canada.
[3]Deltares, Unit Soil and Groundwater Systems, Utrecht, The Netherlands

*Correspondence to*: Edward R. Jones (e.r.jones@uu.nl)



**Abstract.**

Continually improving and affordable wastewater management provides opportunities for both pollution reduction and clean water supply augmentation, whilst simultaneously promoting sustainable development and supporting the transition to a circular economy. This study aims to provide the first comprehensive and consistent global outlook on the state of domestic and industrial wastewater production, collection, treatment and re-use. We use a data-driven approach, collating, cross-examining and standardising country-level wastewater data from online data resources. Where unavailable, data is estimated using multiple linear regression. Country-level wastewater data are subsequently downscaled and validated at 5 arc-minute (~10 km) resolution. This study estimates global wastewater production at 359.5 billion $m^3$ $yr^{-1}$, of which 63% (225.6 billion $m^3$ $yr^{-1}$) is collected and 52% (188.1 billion $m^3$ $yr^{-1}$) is treated. By extension, we estimate that 48% of global wastewater production is released to the environment untreated, which is significantly lower than previous estimates of ~80%. An estimated 40.7 billion $m^3$ $yr^{-1}$ of treated wastewater is intentionally re-used. Substantial differences in per capita wastewater production, collection and treatment are observed across different geographic regions and by level of economic development. For example, just over 16% of the global population in high income countries produce 41% of global wastewater. Treated wastewater re-use is particularly significant in the Middle East and North Africa (15%) and Western Europe (16%), while containing just 5.8% and 5.7% of the global population, respectively. Our database serves as a reference for understanding the global wastewater status and for identifying hotspots where untreated wastewater is released to the environment, which are found particularly in South and Southeast Asia. Importantly, our results also serve as a baseline for evaluating progress towards many policy goals that are both directly and indirectly connected to wastewater management (e.g. SDGs). Our spatially-explicit results available at 5 arc-minute resolution are well suited for supporting more detailed hydrological analyses such as water quality modelling and large-scale water resource assessments, and can be accessed at: https://doi.pangaea.de/10.1594/PANGAEA.918731 (Jones et al., 2020). A temporary link to this dataset for the review process can be accessed at: https://www.pangaea.de/tok/6631ef8746b59999071fa2e692fbc492c97352aa.



## 1. Introduction

Clean water is essential for supporting human livelihoods, achieving sustainable development and maintaining ecosystem health. All major human activities, such as crop and livestock production, manufacturing of goods, power generation and domestic activities rely upon the availability of water in both adequate quantities and of acceptable quality at the point of

intended use (van Vliet et al., 2017; Ercin and Hoekstra, 2014). It is increasingly recognised that conventional water resources such as rainfall, snowmelt and runoff captured in lakes, rivers and aquifers are insufficient to meet human demands in water scarce areas (Jones et al., 2019; Hanasaki et al., 2013; Kummu et al., 2016). Whilst increases in water use efficiencies can somewhat reduce the water demand and supply gap, these approaches must be combined with supply and quality enhancement strategies (Gude, 2017). Conventional supply enhancement strategies, such as reservoir construction, surface water diversion

and pipeline construction are contingent on geographic and climate factors, can face strong public opposition and often lack water quality considerations.

A growing set of viable but unconventional water resources offer enormous potential for narrowing the water demand-supply gap towards a water-secure future. Unconventional water resources encapsulate a range of strategies across different scales,

from localised fog-water and rainwater harvesting, to mega-scale desalination plants and wastewater treatment and re-use facilities (Jones et al., 2019; Morote et al., 2019; Qadir et al., 2020). The use of unconventional water resources has grown rapidly in the last few decades, often out of necessity, and their importance across various geographic scales is already irrefutable (Jones et al., 2019; Qadir et al., 2018). Furthermore, continually improving unconventional water resources technologies have permitted more efficient and economical 'tapping' of water resources which were previously unusable due

to access constraints or the added costs related to unsuitable water quality (e.g. seawater desalination, wastewater treatment).

Wastewater is broadly defined as 'used' water that has been contaminated as a result of human activities (Mateo-Sagasta et al., 2015). Whilst agricultural runoff is rarely collected or treated (WWAP, 2017), return flows from domestic and industrial sources (henceforth 'wastewater') can be collected in infrastructure including piped systems (sewerage) or on-site sanitation

systems (septic tanks; pit latrines). Wastewater is increasingly recognised as a reliable and cost-effective source of freshwater, particularly for agricultural applications (WWAP, 2017; Jiménez and Asano, 2008). Yet, wastewater remains an 'untapped' and 'undervalued' resource (WWAP, 2017). Wastewater treatment improves the quality of 'used' water sources to reduce contaminant levels below sectoral quality thresholds for intentional re-use or to minimise the environmental impacts of wastewater return flows. Treated wastewater flows can also provide a significant source of (clean) freshwater flows for

maintenance of river flows, especially during drought (Luthy et al., 2015). Where treated wastewater discharges form a substantial part of the river flow, de-facto wastewater re-use, defined as the incidental presence of treated wastewater in a water supply, can be high (Rice et al., 2013; Beard et al., 2019). Treated wastewater can also be used for groundwater recharge, helping to preserve the viability of freshwater extraction from groundwater into the future (Qadir et al., 2015), in addition to





applications in agroforestry systems (El Moussaoui et al., 2019) and aquaculture (Khalil and Hussein, 2008). In summary,
wastewater treatment can improve river water quality and ecosystem health, whilst providing an alternative source of
freshwater for human use and subsequently reducing competition for conventional water supplies.

Historically, wastewater (both treated and untreated) has been predominantly used for non-potable purposes, particularly
agriculture and landscape irrigation (Qadir et al., 2007; WWAP, 2017; Zhang and Shen, 2017). Agricultural activities are
expected to increasingly rely on alternative water resources, as this sector has the largest water demands globally (Wada et al.,
2013). Furthermore, the agricultural sector faces reductions in conventional water resources allocation (Sato et al., 2013). The
reliable supply of water, reduced need for additional fertilizer and potential for growing high value vegetables promote
wastewater irrigation in water-scarce developing countries (Sato et al., 2013). However, much of the wastewater currently re-
used is inadequately treated or even untreated (Qadir et al., 2010; Scott et al., 2010). Demands for wastewater are increasing
at a faster pace than treatment solutions and institutions to ensure the safe distribution and management of wastewater (Sato
et al., 2013). The primary challenge in promoting re-use is ensuring safety – both for human and ecosystem health – and thus
ensuring that wastewater is adequately treated prior to use or environmental discharge (WWAP, 2017). This is needed to
achieve the required paradigm shift in water resources management, whereby wastewater is viewed as a resource (for energy,
nutrients and water) rather than as 'waste' (WWAP, 2017; Qadir et al., 2020).


To understand the current state and explore the future potential of wastewater as a resource, a comprehensive and consistent
global assessment of wastewater production, collection, treatment and re-use is required. This information is essential for
assessing progress towards Sustainable Development Goal 6, whereby one of the targets is specifically focused on water quality
improvement by halving the proportion of untreated wastewater and promoting safe re-use globally (SDG6.3). Furthermore,
this information is important for identifying hotspots where improvements in wastewater management are highly necessary
and as input data for a range of scientific assessments (e.g. stream water quality dynamics, water scarcity assessments).
However, the availability of wastewater data at the continental and global scales is sparse, and often outdated or from
inconsistent reporting years (Sato et al., 2013). Previous studies remain limited in their approach and estimates, relying on a
few data sources covering less than half of the countries across the world (Mateo-Sagasta et al., 2015; Sato et al., 2013). A
recent study explored the global and regional 'potential' of wastewater as a water, nutrient and energy source, but did not
address the collection, treatment and re-use aspects of wastewater (Qadir et al., 2020). This paper presents the first global
assessment of spatially-explicit wastewater production, collection, treatment and re-use, consistently combing different data
sources. Country-level quantifications are downscaled to gridded (5 arc-minute resolution) level for inclusion in large-scale
water resource assessments and water quality models.





## 2. Methods

### 2.1 Wastewater Data Sources


The fate of domestic and industrial wastewater after production can follow a number of paths (Figure 1). Wastewater from these activities can be collected, typically in sewers, septic tanks or pit latrines, or uncollected and discharged directly to the environment (e.g. open defecation). Collected wastewater can undergo treatment, ranging from basic primary treatment (removing suspended solids) to specialised tertiary or triple barrier treatment (nutrient removal, toxic compound removal), or can be discharged to the environment untreated (Mateo-Sagasta et al., 2015). When treated, wastewater can be released to the environment or intentionally re-used as a 'fit-for-purpose' water source. Untreated wastewater can similarly be discharged to the environment or intentionally used as a source of freshwater. Furthermore, both treated and untreated wastewater can be used unintentionally where wastewater is incidentally present in a water supply ('de-facto reuse').



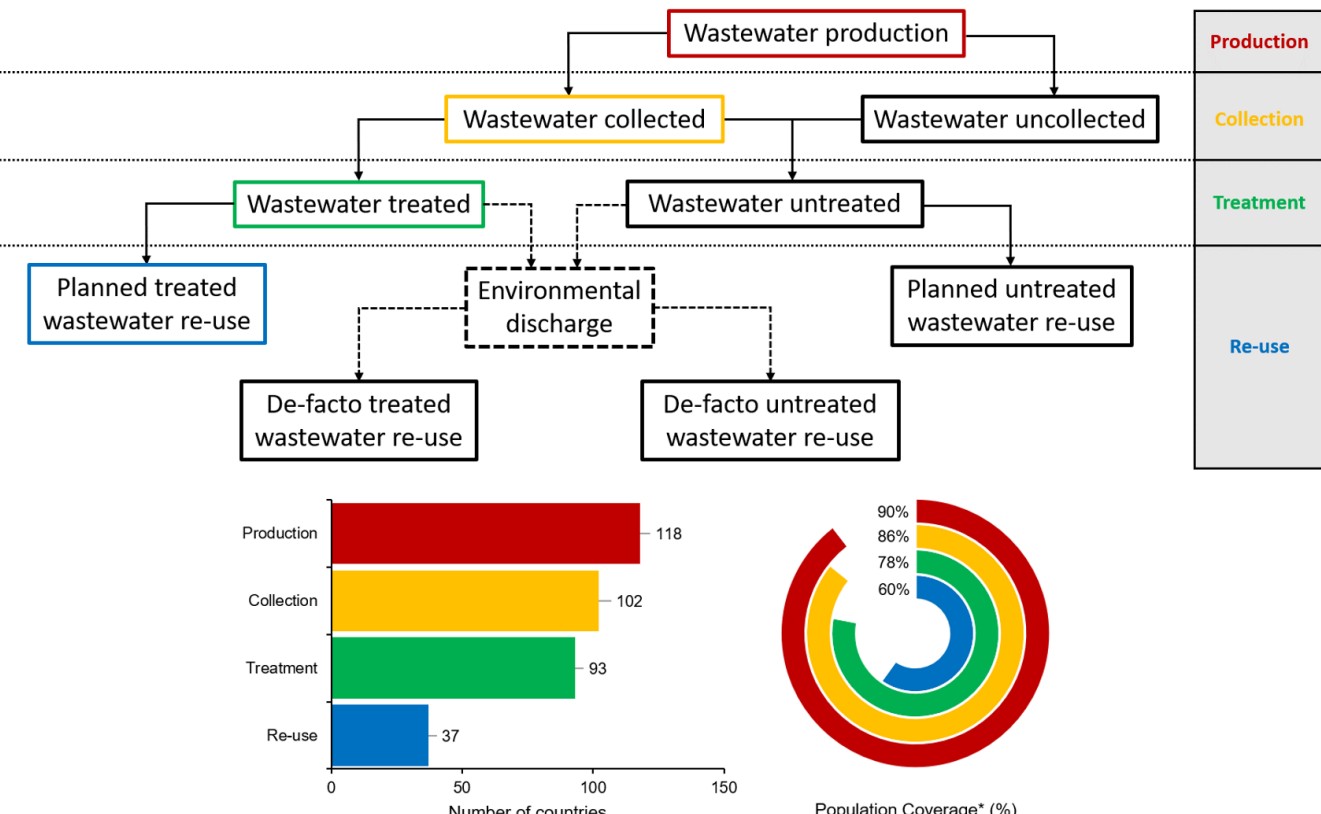

**Fig 1.** Fate of wastewater (a), including wastewater data availability with number of countries for which wastewater data is available (b) and percentage of population coverage (c).

*Population coverage indicates the proportion of the global population for which wastewater data is available.


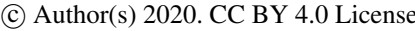



Country-level wastewater data was collated from four online databases: Global Water Intelligence (GWI, 2015); Food and Agricultural Organisation of the United Nations (FAO-AQUASTAT, 2020), Eurostat (Eurostat, 2020) and the United Nations Statistics Division (UNSD, 2020). For consistency, the year 2015 was selected for all wastewater data. Where wastewater data

from the online sources was reported in a different year (up to a maximum of 10 years, i.e. 2006 onwards), all wastewater data was standardised to 2015 based on data from the most recent reporting year (see Table 1 for the standardisation method).

Data from different sources was cross examined to check for consistency and to remove implausible data. Where large discrepancies (>one order of magnitude) existed between different data sources for a country, data points were excluded. For

example, GWI report Kazakhstan to produce 6,205 million $m^3$ $yr^{-1}$, whereas the FAO report just 411 million $m^3$ $yr^{-1}$. Similarly, the FAO report Moldova to produce 46.7 million $m^3$ $yr^{-1}$ compared to 672.1 million $m^3$ $yr^{-1}$ by the UNSD. In total, reported data for 11 countries were excluded for wastewater production. For wastewater collection and treatment, percentage data was cross referenced with reported connection rates (i.e. percentage population connected to wastewater collection/ treatment). Six and seven countries were excluded for collection and treatment, respectively. For example, GWI report a 95.2% collection rate

for Azerbaijan, whilst the UNSD report that only 32.4% of people are connected to wastewater collection systems. Similarly, GWI report only a 17% treatment rate in Hong Kong, whereas the UNSD report that 93.5% of people are connected to wastewater treatment plants. No data points were excluded for wastewater re-use. In a small number of cases where percentage values obtained were marginally illogical (i.e. wastewater collection < wastewater treatment; wastewater treatment < wastewater re-use), percentage values were set to the proceeding level in the wastewater chain (Figure 1).


Table 1 displays the data sources and the associated number of countries with wastewater data for production, collection, treatment and re-use. The procedure for standardising the data to the year 2015, when required, is presented. The methods used to compile wastewater production, collection, treatment and re-use data from multiple sources to provide a single quantification per country is also displayed. Lastly, the population coverage in both percentage terms and by the number of unique countries

is displayed both per geographic region and by economic classification. The number of unique countries and the population coverage of data at each stage of the wastewater chain is also displayed in Figure 1. Both the number of countries and population coverage reduces through the wastewater chain, with available wastewater data decreasing from 118 to 37 countries (90% to 60% population coverage) from wastewater production to wastewater re-use data.




**Table 1.** Wastewater data sources and availability by population coverage and number of unique countries (in square brackets). Method for standardisation of wastewater data to 2015 and the method for compiling wastewater data from

multiple sources into a single quantification per country.

| | Data sources* | Standardisation to 2015 | Data compiling method | Regional aspects | | Economic aspects | |
|---|---|---|---|---|---|---|---|
| | | | | Region | Population coverage ** | Classification | Population coverage*** |
| **Production** | GWI [94] FAO [98] UNSD [23] Eurostat [20] | Divide by GDP ($) in reporting year, multiply with GDP ($) in 2015. | Average of all available sources. | North America | 100% [2] | High | 99.4% [48] |
| | | | | Latin America & Caribbean | 93.9%) [19] | | |
| | | | | Western Europe | 99.8% [19] | Upper middle | 98.0% [34] |
| | | | | Middle East & North Africa | 98.8% [19] | | |
| | | | | Sub-Saharan Africa | 49.6% [17] | Lower middle | 94.6% [31] |
| | | | | Southern Asia | 96.4% [4] | | |
| | | | | Eastern Europe & Central Asia | 89.4% [23] | Low | 13.3% [5] |
| | | | | East Asia & Pacific | 95.3% [15] | | |
| **Collection** | GWI [95] FAO [55] | Divide by GDP per Capita ($/capita) in reporting year, multiple with GDP ($) in 2015. | GWI data prioritised. FAO data if unavailable. | North America | 100% [2] | High | 99.4% [47] |
| | | | | Latin America & Caribbean | 96.7% [20] | | |
| | | | | Western Europe | 99.8% [18] | Upper middle | 97.7% [29] |
| | | | | Middle East & North Africa | 88.3% [17] | | |
| | | | | Sub-Saharan Africa | 61.1% [13] | Lower middle | 81.0% [21] |
| | | | | Southern Asia | 96.4% [4] | | |
| | | | | Eastern Europe & Central Asia | 69.9% [16] | Low | 34.9% [5] |
| | | | | East Asia & Pacific | 83.6% [12] | | |
| **Treatment** | GWI [76] FAO [78] UNSD [21] | Divide by GDP per Capita ($/capita) in reporting year, multiple with GDP ($) in 2015. | GWI data prioritised. FAO or UNSD where unavailable (most recent reporting year prioritised) | North America | 100% [2] | High | 98.4% [46] |
| | | | | Latin America & Caribbean | 90.0% [17] | | |
| | | | | Western Europe | 99.8% [19] | Upper middle | 91.2% [27] |
| | | | | Middle East & North Africa | 65.9% [13] | | |
| | | | | Sub-Saharan Africa | 25.7% [8] | Lower middle | 69.4% [15] |
| | | | | Southern Asia | 95.2% [3] | | |
| | | | | Eastern Europe & Central Asia | 73.4% [21] | Low | 27.1% [5] |
| | | | | East Asia & Pacific | 80.2% [10] | | |
| **Re-use** | GWI [20] FAO [32] | Wastewater production normalised to reporting year of wastewater re-use based on GDP ($), percentage re-use calculated, applied to 2015 production data. | GWI data prioritised. FAO data if unavailable. | North America | 90.0% [1] | High | 68.7% [19] |
| | | | | Latin America & Caribbean | 67.2% [5] | | |
| | | | | Western Europe | 42.5% [3] | Upper middle | 77.7% [10] |
| | | | | Middle East & North Africa | 83.0% [13] | | |
| | | | | Sub-Saharan Africa | 21.5% [6] | Lower middle | 48.7% [4] |
| | | | | Southern Asia | 74.9% [1] | | |
| | | | | Eastern Europe & Central Asia | 0.6% [2] | Low | 24.8% [4] |



* Abbreviations for the data sources are as follows: Global Water Intelligence (GWI), Food and Agricultural Organisation of the United Nations (FAO), United Nations Statistics Department (UNSD), European Union Statistics Office (Eurostat). The number of countries per data source is displayed in square brackets.

**Data availability per region expressed as a percentage of the total population, with the number of countries in square brackets. Total number of countries per geographic region are: East Asia and Pacific [38], Eastern Europe and Central Asia [30], Latin America and Caribbean [41], Middle East and North Africa [21], North America [3], Southern Asia [8], Sub-Saharan Africa [48] and Western Europe [26].

***Data availability per economic classification expressed as a percentage of the total population, with the number of countries in square brackets. Total number of countries per economic classification are: High [76], Upper middle [56], Lower middle [52], and Low [31] income.

## 2.2 Regression for country-level predictions

Datasets of predictor variables for regression analyses were downloaded from multiple sources (see overview Table 2). Datasets spanned a wide range of predictor variables covering social (e.g. total and urban population), economic (e.g. GDP, Human Development Index), hydrological (e.g. irrigation water scarcity) and geographic (e.g. land area, agricultural land) dimensions. The selected predictor variables were expected to have a physical basis for correlation with wastewater production,

collection, treatment or re-use. Where appropriate, datasets from these sources were combined to produce comparable metrics for countries of different population and geographic sizes (e.g. GDP per Capita [$ per capita]; desalination capacity per capita [$m^3 \ yr^{-1}$ per capita]). Values were taken for the year 2015, where available, or from the closest reporting year when unavailable. Irrigation water scarcity and desalination capacity were taken from 2019 and 2018, respectively. Data was transformed, as appropriate, to ensure normality.








**Table 2.** Data sources of predictor variables for wastewater production, collection, treatment and re-use regression analysis.

| Data Source | Predictor Variable | Year | Link |
|---|---|---|---|
| **World Bank** | Land area (km$^2$) | 2015 | https://data.worldbank.org/indicator/AG.LND.TOTL.K2 |
| | Total population (millions) | 2015 | https://data.worldbank.org/indicator/sp.pop.totl |
| | Urban population (%) | 2015 | https://data.worldbank.org/indicator/SP.URB.TOTL |
| | GDP (billion US$) | 2015 | https://data.worldbank.org/indicator/NY.GDP.MKTP.CD |
| | Access to basic sanitation (%) | 2015 | https://data.worldbank.org/indicator/SH.STA.BASS.ZS |
| | Mortality rate attributed to unsafe WASH | 2015 | https://data.worldbank.org/indicator/SH.STA.WASH.P5 |
| | Access to internet (%) | 2015 | https://data.worldbank.org/indicator/it.net.user.zs |
| | Access to electricity (%) | 2015 | https://data.worldbank.org/indicator/EG.ELC.ACCS.ZS |
| | People practicing open defecation (%) | 2015 | https://data.worldbank.org/indicator/SH.STA.ODFC.ZS |
| | Agricultural Land (%) | 2015 | https://data.worldbank.org/indicator/AG.LND.AGRI.ZS |
| | Fertilizer consumption (kg /ha arable land) | 2015 | https://data.worldbank.org/indicator/ag.con.fert.zs |
| | Renewable internal water resources (billion m$^3$) | 2015 | https://data.worldbank.org/indicator/ER.H2O.INTR.K3 |
| **United Nations Development Programme** | Human Development Index (-) | 2015 | https://dasl.datadescription.com/datafile/hdi-2015/ |
| **World Resources Institute** | Baseline Irrigation Water Scarcity (-) | 2019 | https://www.wri.org/resources/data-sets/aqueduct-30-country-rankings |
| **Global Water Intelligence** | Desalination capacity (million m$^3$ yr$^{-1}$) | 2018 | https://www.desaldata.com/ as synthesised in Jones et al. (2019) |

Multiple linear regression was used to predict country-level wastewater variables (production, collection, treatment and re-use) for countries without reported data. Stepwise elimination was used for feature selection to remove redundant predictor variables and reduce overfitting. Wastewater production was predicted in volumetric flow rate units (million m$^3$ yr$^{-1}$). Conversely, wastewater collection, treatment and re-use were predicted as a percentage of wastewater production. Predicted values of percentages were bounded to the 0 – 100% range (i.e. <0 = 0; >100 = 100). Predicted percentages were subsequently applied to reported or predicted values of wastewater production to obtain wastewater collection, treatment and re-use in volumetric flow rate units. Bootstrap regression was used to quantify the uncertainty in the predictions (by geographic region, economic classification and at the global scale) at the 95th confidence level. In total, 1,000 regressions with random sampling and replacement were fit to provide predictions at countries lacking data. Wastewater observations were combined with these 1,000 bootstrapped predictions, with the 2.5th and 97.5th confidence intervals taken as lower and upper confidence limits, respectively.





Wastewater data (reported and predicted) are at the national level, for the 215 countries as listed by the World Bank (https://data.worldbank.org/country). Wastewater data are also aggregated to eight geographic regions: 1) East Asia & Pacific; 2) Eastern Europe & Central Asia; 3) Latin America & Caribbean; 4) Middle East & North Africa; 5) North America; 6) Southern Asia; 7) Sub-Saharan Africa; and 8) Western Europe. Furthermore, data is also aggregated to four economic classifications based on the World Bank Atlas Method: 1) High income (>$12,056 GNI per capita); 2) Upper middle income

($3896 to $12,055); 3) Lower middle income ($966 to $3895); and 4) Low income (<$995). Predicted wastewater data was used to supplement reported data, where unavailable, to develop a comprehensive global outlook of wastewater production, collection, treatment and re-use.

### 2.3  Downscaling and validation

Country-level wastewater production, collection, treatment and re-use data was downscaled to 5 arc-minute resolution (~10km at the equator) based on averaged sum of annual domestic and industrial return flow data (henceforth 'return flow') from the PCRaster GLOBal Water Balance Model for the years 2006 - 2015 (PCR-GLOBWB 2; (Sutanudjaja et al., 2018)). Return flows represent the water extracted for a specific sectoral purpose, but is not consumed, and hence returns to and dynamically interacts with surface and ground water hydrology (de Graaf et al., 2014; Sutanudjaja et al., 2018). Domestic return flows only

occurs where the urban and rural population have access to water, whereas industrial return flow occurs from all areas where water is withdrawn (Wada et al., 2014). Both domestic and industrial return flows are dependent on country-specific recycling ratios based on GDP and the level of economic development (Wada et al., 2011). Grid cell return flow was divided by the country's total return flow to obtain the fraction per grid cell. Wastewater production was downscaled directly proportionally to return flows by multiplying the grid cell return flow fraction per grid cell with wastewater production at the country level.


For wastewater collection, treatment and re-use, minimum return flow criteria were set. These criteria prevent allocation of collection, treatment and re-use to grid cells with low levels of municipal activities, where central collection and treatment is not feasible economically. Furthermore, these criteria increase allocation to grid cells with high populations and industrial activities, where wastewater facilities are more extensive. Minimum thresholds per country were determined based on quantiles

related to the overall wastewater collection and treatment in a country. For example, if a country's wastewater collection is 70% (or a ratio of 0.7) and treatment 50% (0.5), the grid cell return flow quantiles are set to 1 – Ratio (i.e. 0.3 and 0.5, respectively). These quantiles are applied to all grid cells with wastewater production within a country to determine a minimum threshold for allocation. Grid cells not meeting the threshold were excluded, and an adjusted allocation fraction based only on grid cells that meet the criteria was calculated. Wastewater re-use was downscaled using the same return flow quantiles as for

wastewater treatment, with an additional criterion to account for water scarcity, a key driver of wastewater re-use. The ratio of water withdrawals to water availability in a grid cell was used as the indicator for water scarcity, as per convention, with the





commonly used ratio of 0.2 set as the water scarcity threshold (Hanasaki et al., 2018; Liu et al., 2017). The adjusted fraction was multiplied by country-level wastewater collection, treatment and re-use data to downscale values per grid cell.

The location and design capacity of individual wastewater treatment plants were used to validate the downscaled wastewater treatment data. Reported data for 25,901 wastewater treatment plants located across Europe were obtained from the European Environmental Agency (EEA, 2019). Data for a further 4,283 wastewater treatment plants was obtained for the contiguous United States from the US Environmental Protection Agency (US-EPA, 2020). An additional 478 wastewater treatment plants, distributed globally, were extracted from the GWI wastewater database (GWI, 2015). For EEA and GWI wastewater treatment

plants, treatment capacity reported only in Population Equivalent (PE) was approximated in volume flow rate units based on the linear regression obtained for wastewater treatment plants reporting both capacity in both population equivalent and volume flow rate (EEA: $R^2 = 0.80$, $p < 0.001$; GWI: $R^2 = 0.81$, $p < 0.001$). Wastewater treatment plants were assigned to their nearest grid cell and treatment capacities were aggregated. In total, wastewater treatment data was available for 22,133 unique grid cells. For validating downscaled wastewater re-use, only plants (with treatment capacity > 1 million $m^3$ $yr^{-1}$) using tertiary or

higher wastewater treatment technologies were considered. In total, 572 wastewater treatment plants in the EEA database met this criterion. A further 78 wastewater treatment plants, which are specifically designated as wastewater re-use facilities, were sourced from the GWI database. In total, wastewater re-use data was available for 601 grid cells. Downscaled wastewater treatment and re-use was compared to wastewater design capacities.

To account for the large variation in the treatment capacities of wastewater treatment plants considered, in addition to the geographical mismatch between where wastewater is produced and treated (i.e. wastewater treatment plants are typically located on the outskirts of urban areas), validation occurred at differing geographical scales. Wastewater treatment plant capacity was divided by wastewater production per capita to approximate the number of people that the wastewater treatment plant serves. If the population served by a wastewater plant exceeds the grid cell population, the validation extent is expanded

to the directly neighbouring cells. This is allowed to occur until the population served by the treatment plant is reached, but up to a maximum of 3 iterations, reflecting a radius of ~30km around the wastewater treatment plant. The total downscaled wastewater treated over the extended area is then compared to that of the treatment plant.

   To quantify the performance of the downscaling approaches, the root-mean-square error (RMSE) and mean bias (BIAS) were

calculated. Normalised values of RMSE and BIAS were calculated (nRMSE and nBIAS) by dividing by the standard deviation of the wastewater treatment plant capacity. Pearson's (r) coefficients was calculated to quantify the linear dependence, with $R^2$ values based on both the linear and log-log relationship between downscaled and observed values also calculated.



## 3. Results

### 3.1 Regression and country-level predictions

**Table 3.** Wastewater production, collection, treatment and re-use multiple linear regression results.

| Regression model | Explanatory Variables (units) | *B* (SE *B*) | β | P | Adjusted R$^2$ |
|---|---|---|---|---|---|
| **Production (log)** | Intercept (-) | -1.68 (0.45) | | ** | 0.89** |
| | GDP (log $ yr$^{-1}$ per capita) | 0.45 (0.06) | 0.31 | ** | |
| | Population (log millions) | 1.02 (0.03) | 0.96 | ** | |
| | Access to basic sanitation (%) | 0.02 (0.00) | 0.19 | ** | |
| **Collection** | Intercept (-) | -80.73 (11.06) | | ** | 0.69** |
| | Human development index (-) | 120.82 (26.94) | 0.50 | ** | |
| | Urban population (%) | 0.22 (0.13) | 0.14 | . | |
| | Wastewater production (log m$^3$ yr$^{-1}$ per capita) | 8.01 (2.97) | 0.25 | * | |
| **Treatment** | Intercept (-) | -61.32 (14.06) | | * | 0.80** |
| | Wastewater collection (%) | 0.72 (0.08) | 0.66 | * | |
| | GDP (log $ yr$^{-1}$ per capita) | 7.2 (1.88) | 0.28 | * | |
| **Re-use (primary)** | Intercept (-) | -5.29 (4.59) | | 0.26 | 0.70** |
| | Desalination capacity (sqrt m$^3$ yr$^{-1}$ per capita) | 1.50 (0.78) | 0.29 | . | |
| | Treated wastewater for irrigation water scarcity alleviation (-) | 13.66 (3.50) | 0.60 | * | |
| **Re-use (alternate)** | Intercept (-) | -4.11 (6.10) | | 0.50 | 0.61** |
| | Desalination capacity (sqrt m$^3$ yr$^{-1}$ per capita) | 3.22 (0.63) | 0.63 | * | |
| | Treated wastewater (%) | 0.23 (0.12) | 0.24 | . | |

*B* indicates unstandardised regression weights, SE *B* indicates the standard error of *B*, β indicates standardised regression weights. Significance level represent by: '**' (p<0.001), '*' (p<0.01), '.' (p<0.1), or as stated numerically.

The results of the regression analysis for wastewater production, collection, treatment and re-use are summarised in Table 3. All regression models were significant at the p<0.001 level with adjusted R$^2$ values ranging between 0.61 and 0.89. Country-level observed versus simulated wastewater production (log million m$^3$ yr$^{-1}$), collection (%), treatment (%) and re-use (%) data are displayed in Figure 2. The regression equations were applied for 97, 113, 122 and 178 countries with no or excluded data representing 10%, 14%, 22% and 40% of the global population for wastewater production, collection, treatment and re-use, respectively.

Wastewater production was best predicted based on total population, GDP per capita and access to basic sanitation. A significant regression equation was found (p<0.001) with an adjusted R$^2$ value of 0.89, with all predictor variables also significant at the p<0.001 level. Whilst the number of people within a country is the strongest influence over total wastewater production (β = 0.96), the average economic output per inhabitant (β = 0.31) and the level of access to wastewater services (β = 0.19), such as flushing toilets to piped sewers are important for determining the amount of wastewater produced per capita. These three factors therefore account for the combined effect of population size and variations in wastewater production per capita linked to economic and development factors in determining total wastewater production in a country. Comparing observed with predicted total wastewater production data demonstrates the overriding importance of a country's population, with wastewater production spread across multiple orders of magnitude for countries irrespective of geographical region or economic classification (Figure 2a).





Wastewater collection was predicted (adjusted $R^2$ = 0.69; p <0.001) based on the human development index (HDI), urban
population and wastewater production per capita. HDI, an overarching proxy for level of development, was found to be the
strongest influence over wastewater collection (β = 0.50; p<0.001). Urban population (β = 0.14; p<0.01) and wastewater
production per capita (β = 0.25; p<0.01) were also significant but less important predictor variables of wastewater collection.
For urban population, a greater proportion of a population living in urban areas resulted in higher collection rates for the
country, while higher levels of wastewater production per capita corresponded to larger collection rates. The observed versus
predicted wastewater collection rates are depicted in Figure 2b, which displays the trend across different geographic zones and
economic classifications.

Wastewater treatment was predicted (adjusted $R^2$ = 0.80; p <0.001) based on GDP per capita (β = 0.28; p < 0.01) and
wastewater collection (β = 0.66; p < 0.01). Countries with larger economic outputs per capita likely have more resources for
wastewater treatment, resulting in higher overall treatment rates. As wastewater treatment is dependent upon wastewater
collection, countries with higher wastewater collection rates typically also treat a greater proportion of their wastewater.
Observed versus predicted wastewater treatment rates are displayed in Figure 2c.

The amount of wastewater treated will determine the maximum potential for treated wastewater re-use within a country. Water
scarcity, particularly when driven by high irrigation water demands, is also a primary driver of wastewater re-use (Garcia and
Pargament, 2015). To account for this relationship, the fraction of wastewater undergoing treatment processes and irrigation
water scarcity was multiplied to give an integrated metric indicating the 'availability of treated wastewater for irrigation water
scarcity alleviation'. Wastewater re-use was predicted (adjusted $R^2$ = 0.70; p <0.001) from this metric (β = 0.60; p < 0.01) in
combination with the desalination capacity per capita (β = 0.29; p < 0.1), as an indicator of the prevalence of unconventional
water resources in a country. The observed versus predicted wastewater re-use rates from this regression are displayed in
Figure 2d. Irrigation water scarcity data was unavailable for 53 countries, mostly small island nation. Here an alternate
regression model was constructed based on desalination capacity per capita (β = 0.63; p < 0.01) and wastewater treatment (β
= 0.24; p < 0.1) only, resulting in a slightly lower explained variance ($R^2$=0.61). While these countries represent <1% of the
global population, this alternate regression was necessary to account for wastewater re-use occurring particularly in water-
scarce small island nations. These islands typically lack renewable water resources and hence unconventional water resources
such as desalinated water and treated wastewater represent a significant proportion of the water availability (Jones et al., 2019).

**Fig 2.** Observed versus predicted wastewater production (a), collection (b), treatment (c) and re-use (d) from regression

320                                                    analysis.




### 3.2 Global wastewater production, collection, treatment and re-use

Globally, this study estimates that 359.4 (358.0 – 361.4) billion m$^3$ of wastewater is produced annually, with a global average of 49.0 (48.8 – 49.2) m$^3$ yr$^{-1}$ per capita. Global wastewater collection and treatment is estimated at 225.6 (224.4 – 226.9) and 188.1 (186.6 – 189.3) billion m$^3$ yr$^{-1}$, respectively. These values indicate that approximately 63% and 52% of globally produced

wastewater is collected and treated, respectively, with approximately 84% of collected wastewater undergoing a treatment process. Wastewater re-use is estimated a 40.7 billion (37.2 – 47.0) m$^3$ yr$^{-1}$, representing approximately 11% of the total volume of wastewater produced. This estimate also indicates that approximately 22% of treated wastewater undergoes intentional re-use, with the remaining 78% (totalling 147.4 billion m$^3$ yr$^{-1}$) discharged to the environment. This compares to the estimated 171.3 billion m$^3$ yr$^{-1}$ of wastewater discharged directly to the environment without undergoing any form of treatment. It is

worth highlighting that the vast majority of wastewater data is from reported sources, with just 2.4%, 4.8% and 5.2% of global wastewater production, collection and treatment from predicted values using regression. This occurs both due to the high population coverage and due to the missing data primarily being from developing nations, where wastewater production per capita and percentage collection and treatment rates are lower. The global quantification of wastewater re-use relies more heavily on predicted values, constituting 23.4% of re-use volume globally. This occurs primarily due to poor data availability,

particularly in countries with large populations in Eastern Europe and Central Asia (e.g. Russia, Turkey and Poland) and Western European nations where wastewater treatment rates are generally high but the proportion of wastewater re-used relies on simulations (e.g. Germany, Italy and Greece).

Table 4 displays wastewater production per capita (m$^3$ yr$^{-1}$ per capita) and wastewater production, collection, treatment and

re-use (billion m$^3$ yr$^{-1}$), aggregated from the country data (reported + simulated) at the global scale and by region and level of economic development. Figure 3 displays wastewater data plotted at the country scale in proportional terms (m$^3$ yr$^{-1}$ per capita for production; percentage of produced wastewater for collection, treatment and re-use), facilitating direct comparisons between countries.

Substantial differences in wastewater production, collection, treatment and re-use occur across different geographic regions and by the level of economic development. Wastewater production per capita is notably highest in North America at 209.5 m$^3$ yr$^{-1}$ per capita, over double that of Western Europe (91.7 m$^3$ yr$^{-1}$ per capita), the next highest producing region per capita. When considering individual countries in these regions, the USA (211 m$^3$ yr$^{-1}$ per capita) and Canada (198 m$^3$ yr$^{-1}$ per capita), in addition to small, prosperous European countries (e.g. Andorra: 257 m$^3$ yr$^{-1}$ per capita; Austria: 220 m$^3$ yr$^{-1}$ per capita; Monaco:

203 m$^3$ yr$^{-1}$ per capita) are the highest producers per Capita. Comparatively, the larger Western European countries have lower wastewater production per capita, with Germany, the U.K. and France at 92, 92 and 66 m$^3$ yr$^{-1}$ per capita, respectively. Conversely, most Sub-Saharan African nations produce less than 10 m$^3$ yr$^{-1}$ per capita. Wastewater production values are comparable to the World Health Organisations absolute minimum water requirements for survival of 2.7 m$^3$ yr$^{-1}$ per capita





(WHO, 2011) in countries such as Niger (2.7 m³ yr⁻¹ per capita), Burkina Faso (3.4 m³ yr⁻¹) and Ethiopia (4.2 m³ yr⁻¹ per
capita). Aggregated for the region, Sub-Saharan Africa produces approximately 20 times less wastewater than North America
per capita, at 11.0 m³ yr⁻¹ per capita.

In volumetric flow rate terms, East Asia and Pacific produces the most wastewater (117.6 billion m³ yr⁻¹), coinciding with the
largest population share (~31%). Conversely, Southern Asia produces just ~7% of global wastewater despite a population
share of ~24%, whereas the ~5% of people living in North America account for ~20% of global wastewater production.
Wastewater production also varies greatly with level of economic development. The prominent discrepancies between
economic classifications indicates a strong relationship between wealth and wastewater production regardless of geographic
location. Wastewater production per capita more than doubles at each income classification level from low income (6.4 m³ yr⁻¹
per capita) to high income (126.0 m³ yr⁻¹ per capita). With respect to population size, people living in high income countries
(~16% global population) produce ~42% of global wastewater, compared to low and lower-middle income countries (~50%
global population) producing ~20% of global wastewater.

Wastewater collection and treatment rates are highest in Western Europe (88% and 86%, respectively) and lowest in Southern
Asia (31% and 16%, respectively) and Sub-Saharan Africa (23% and 16%, respectively). Wastewater collection is notably low
in the East Asia and Pacific region, where total wastewater production is high. Conversely, wastewater collection in the Middle
East and North Africa region is relatively high at 74%, likely resulting from the lack of renewable water supplies. Wastewater
treatment percentages follow similar regional patterns. Notably, wastewater treatment is significantly lower than wastewater
collection in the Latin America and Caribbean and Southern Asia regions, potentially indicative of high rates of untreated
wastewater re-use in these regions. Wastewater collection and treatment percentages follow similar patterns as wastewater
production with respect to income level, with high income countries collecting and treating the majority of their wastewater
(82% and 74%, respectively) down to low income countries with small collection and treatment rates (9% and 4%,
respectively). The proportion of collected wastewater being treated also decreases with income level, at 91%, 73%, 60% and
47% for high, upper middle, lower middle and low income classifications, respectively. The fact that 40% and 53% of collected
wastewater is untreated in the lower middle and low income classifications respectively may also be indicative of the higher
prevalence of intentional untreated wastewater re-use (whereby collected wastewater is re-used without undergoing treatment).

High utilisation of treated wastewater re-use occurs predominantly in the Middle East and North Africa, with the United Arab
Emirates, Kuwait and Qatar re-using more than 80% of their produced wastewater. Water scarce small island developed
nations, including the Cayman Islands, US Virgin Islands and Malta also have high rates of intentional treated wastewater re-
use of 78%, 75% and 67%, respectively. Treated wastewater re-use is prohibitively low in areas with low wastewater treatment
rates, such as Sub-Saharan Africa and Southern Asia. In addition, treated wastewater re-use is also low in areas with sufficient
availability of conventional water resources such as across Scandinavia (where re-use is <5%).



In volumetric flow rate terms, intentional treated wastewater re-use is estimated to be largest in East Asia & Pacific (11.9

billion m$^3$ yr$^{-1}$) and North America (9.1 billion m$^3$ yr$^{-1}$) and lowest in Southern Asia (0.5 billion m$^3$ yr$^{-1}$) and Sub-Saharan

Africa (1.6 billion m$^3$ yr$^{-1}$). Conversely the Middle East & North Africa (27.8%) and Western Europe (17.5%) regions dominate

in percentage terms. In volumetric flow rate units, the Middle East & North Africa (15%) and Western Europe (16%) account

for almost a third of treated wastewater re-use globally, despite only accounting for 5.8% and 5.7% of the global population,

respectively. Approximately half (52%) of intentional treated wastewater re-use occurs in high income countries, with 37%

from upper middle income countries. Intentional treated wastewater re-use is contingent upon the availability of treated

wastewater resources, which is typically more prevalent in high income countries (who both produce more wastewater per

capita and treated a higher percentage of the resource). However, the proportion of treated wastewater intentionally re-used is

higher in the upper middle (25%) and lower middle (25%) income groups than by the high income group (19%).

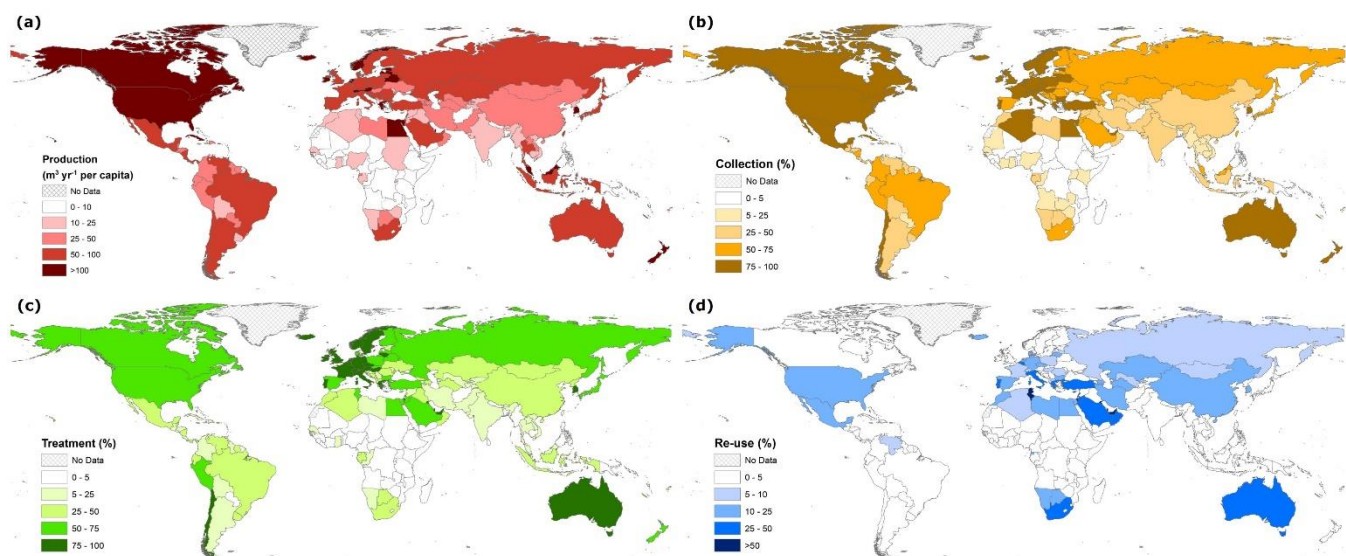


**Fig 3.** Wastewater production (m$^3$ yr$^{-1}$ per capita) (a), collection (%) (b), treatment (%) (c) and re-use (%) (d) at the country

scale.




**Table 4.** Global, regional and level of economic development analysis of wastewater production, collection, treatment and re-use (billion m³ yr⁻¹). The numbers in brackets display the prediction uncertainty (2.5th and 97.5th confidence limits) on the totals based on the results of 1,000 bootstrap regression with random sampling and replacement.


| | Global Population (%) | Production (m³ yr⁻¹ per capita) | Production (billion m³ yr⁻¹) | Collection (billion m³ yr⁻¹) | Treatment (billion m³ yr⁻¹) | Re-use (billion m³ yr⁻¹) |
|---|---|---|---|---|---|---|
| Global | 100 | 49.0 *(48.8 – 49.2)* | 359.4 *(358.0 – 361.4)* | 225.6 *(224.4 – 226.9)* | 188.1 *(186.6 – 189.3)* | 40.7 *(37.2 – 47.0)* |
| **Geographic Region** | | | | | | |
| North America | 4.9 | 209.5 *(209.5 - 209.5)* | 74.7 *(74.7 – 74.7)* | 59.1 *(59.1 – 59.1)* | 50.6 *(50.6 – 50.6)* | 9.1 *(8.8 – 9.5)* |
| Latin America & Caribbean | 8.5 | 67.6 *(67.3 – 67.9)* | 42.1 *(41.9 – 42.3)* | 25.2 *(25.2 – 25.2)* | 15.4 *(15.2 – 15.5)* | 2.1 *(2.0 – 2.5)* |
| Western Europe | 5.7 | 91.7 *(91.7 – 91.8)* | 38.5 *(38.4 – 38.5)* | 33.7 *(33.7 – 33.7)* | 33.0 *(33.0 – 33.0)* | 6.7 *(4.1 – 9.5)* |
| Middle East & North Africa | 5.8 | 51.4 *(51.3 – 51.5)* | 21.9 *(21.8 – 21.9)* | 16.1 *(16.1 – 16.2)* | 11.4 *(11.2 – 11.5)* | 6.1 *(6.0 – 6.2)* |
| Sub-Saharan Africa | 13.6 | 11.0 *(10.1 – 12.4)* | 11.0 *(10.1 – 12.4)* | 2.5 *(2.5 – 2.6)* | 1.8 *(1.7 – 1.9)* | 1.6 *(1.6 – 1.8)* |
| Southern Asia | 23.8 | 14.6 *(14.5 – 14.7)* | 25.6 *(25.4 – 25.7)* | 7.8 *(7.8 – 7.8)* | 4.0 *(4.0 – 4.1)* | 0.5 *(0.5 – 0.8)* |
| Eastern Europe & Central Asia | 6.6 | 57.9 *(57.2 – 58.8)* | 28.2 *(27.8 – 28.6)* | 18.4 *(18.2 – 18.7)* | 14.9 *(14.7 – 15.1)* | 2.6 *(1.3 – 4.4)* |
| East Asia & Pacific | 31.1 | 51.5 *(51.5 – 51.7)* | 117.6 *(117.3 – 117.9)* | 62.8 *(61.9 – 63.8)* | 57.0 *(56.1 – 57.8)* | 11.9 *(11.7 – 13.5)* |
| **Economic Classification** | | | | | | |
| High | 16.1 | 126.0 *(125.9 – 126.2)* | 149.1 *(149.0 – 149.3)* | 121.7 *(121.6 – 121.7)* | 110.4 *(110.4 – 110.5)* | 21.2 *(19.1 – 24.9)* |
| Upper middle | 34.8 | 54.7 *(54.5 – 54.8)* | 139.5 *(139.1 – 139.9)* | 74.8 *(74.6 – 74.9)* | 60.2 *(59.7 – 60.6)* | 15.1 *(13.9 – 16.9)* |
| Lower middle | 40.5 | 22.5 *(22.3 – 22.6)* | 66.8 *(66.4 – 67.4)* | 28.8 *(27.7 – 29.9)* | 17.3 *(16.2 – 18.2)* | 4.4 *(3.6 – 5.7)* |
| Low | 8.6 | 6.4 *(5.0 – 8.5)* | 4.0 *(3.2 – 5.3)* | 0.4 *(0.3 – 0.4)* | 0.2 *(0.1 – 0.2)* | 0.0 *(0.0 – 0.1)* |

### 3.3 Gridded wastewater production, collection, treatment and re-use

Figure 4 displays gridded wastewater production, collection, treatment and re-use, allowing for the identification of hotspot regions and zones at 5 arc-min resolution. Wastewater production occurs across the globe, with hotspots coinciding with the largest metropolitan areas (e.g. Tokyo and Mumbai) where the largest concentration of domestic and industrial activities occurs (Figure 4a). In contrast, wastewater production is close to zero in world regions with low concentrations of people and industrial activities, such as the Sahara desert, inland Australia and the high latitude climate zones (e.g. Northern Canada and

Russia). In countries where municipal activities are heavily concentrated in a small number of cities, such as in the Middle East and Australia, small clusters of grid cells with very high wastewater production (>5 million m³/year) occurs. Wastewater

collection (Figure 4b) and treatment (Figure 4c) is typically more concentrated in urban areas within individual countries. This is particularly prominent in South America and Sub-Saharan Africa. Conversely, downscaled wastewater collection and treatment reflect wastewater production in regions where wastewater collection and treatment rates are very high, such as

Western Europe and Scandinavia. Wastewater re-use is constrained to the lowest area (number of grid cells), being concentrated in regions with only available wastewater resources and conditions of water scarcity.

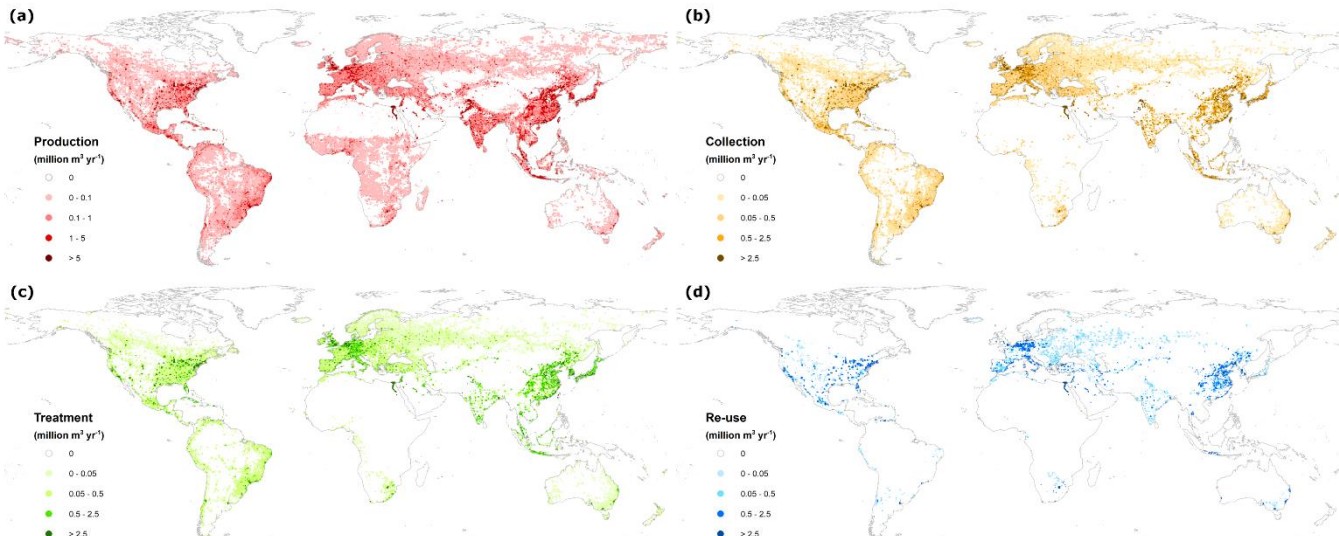

**Fig 4.** Gridded wastewater production (a), collection (b), treatment (c) and re-use (d) (million $m^3$ $yr^{-1}$) at 5 arc-minute spatial
430                                                                                 resolution.

Figure 5a displays the global distribution of the wastewater treatment plants and designated wastewater re-use sites considered in this study. Plant capacities were compared to downscaled quantifications for validation of wastewater treatment (Figure 5b) and wastewater re-use (Figure 5c). Overall, a reasonable performance is obtained at most wastewater treatment and re-use

plants with linear $R^2$ values of 0.58 ($p < 0.001$) and 0.57 ($p < 0.001$), respectively. The observed negative normalised biases suggest that downscaled wastewater treatment (-0.38) and re-use (-0.73) was underestimated with respect to the observed treatment capacities. This may occur due to discrepancies between the design (i.e. maximum) capacity of wastewater treatment plants, which is commonly the capacity that is reported, versus the actual treated wastewater volumes. Factors such as the construction year of wastewater treatment plant are important, as plants are constructed to be larger than current requirements

in anticipation of future increases in wastewater flows.

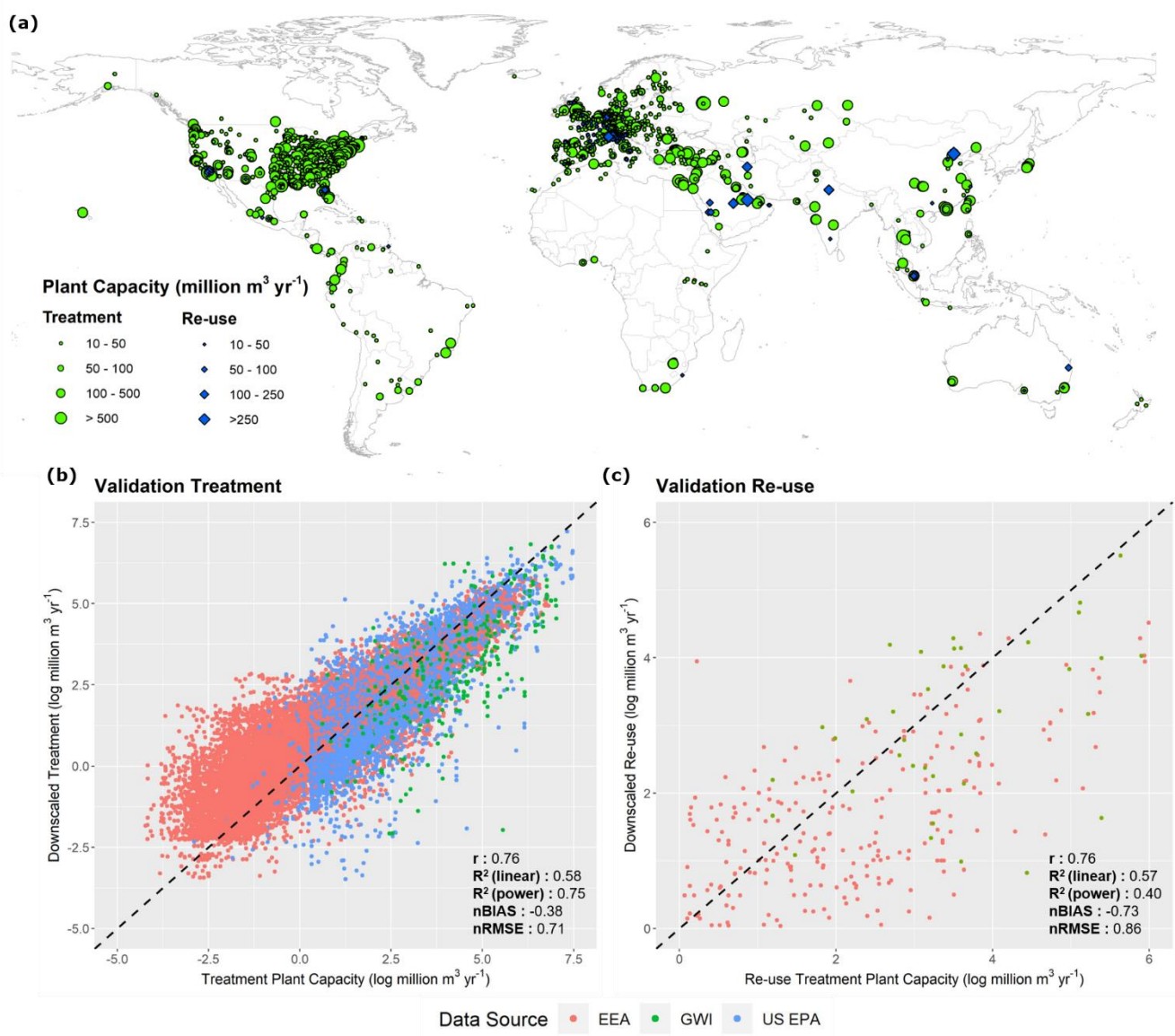

**Fig 5.** Global distribution of wastewater treatment plants and designated wastewater re-use sites (a), and validation of downscaling approach for wastewater treatment (b) and wastewater re-use (c).

**4.   Data availability**

The spatially-explicit wastewater production, collection, treatment and re-use datasets at 5 arc-minutes can be accessed at: https://doi.pangaea.de/10.1594/PANGAEA.918731 (Jones et al., 2020). A temporary link to this dataset for the review process can be accessed at: https://www.pangaea.de/tok/6631ef8746b59999071fa2e692fbc492c97352aa.



## 5. Discussion and conclusions

This study aimed to develop a consistent and comprehensive spatially-explicit assessment of global wastewater production, collection, treatment and re-use for the reference year of 2015. Multiple linear regression models using a diverse set of social, economic, geographic and hydrological datasets were fit for country level wastewater data collated for a variety of sources. These relationships applied for predictions of wastewater production, collection, treatment and re-use for countries where data was unavailable. Bootstrapping with random sampling and replacement was employed to quantify prediction uncertainty. It should be noted that bootstrapping only accounts for uncertainty in the regression terms, not for uncertainties in the underlying source data. Uncertainties associated with wastewater observations are not accounted for in this study, despite likely being substantial. Nevertheless, this study represents the first attempt to simultaneously analyse wastewater production, collection, treatment and re-use for all countries across the globe.

Our global quantification of wastewater production of 359.4 (358.0 – 361.4) billion $m^3$ $yr^{-1}$ is broadly in accordance with previous quantifications, such as 380 billion $m^3$ $yr^{-1}$ quantified based on urban population only (Qadir et al., 2020), and 450 billion $m^3$ $yr^{-1}$ quantified by modelling of return flows in WaterGAP3 (Flörke et al., 2013). Few studies were found analysing the global state of wastewater collection, treatment and re-use. Our quantification of wastewater collection, which is estimated at 225.6 (224.4 – 226.9) billion $m^3$ $yr^{-1}$, can give an important indication of the amount of collected wastewater that goes untreated. At the global scale, this study estimates that wastewater treatment is 188.1 (186.6 – 189.3) billion $m^3$ $yr^{-1}$, or 52% of the produced wastewater. By extension, 48% of produced wastewater is released to the environment without treatment (either directly, or following collection). This is significantly lower than the commonly cited statistic that ~80% of global wastewater is released to the environment without treatment (WWAP, 2012; UN-Water 2015a; UNESCO, 2017). Our quantifications of wastewater treatment must be treated with caution however – particularly in the developing world – as wastewater treatment plants typically operate at capacities below the installed (and usually reported) capacities (Mateo-Sagasta et al., 2015; Murray and Drechsel, 2011) upon which country-level estimates incorporate. Similarly, wastewater plants may be entirely non-functional (mothballed) due to lack of funding and maintenance, or have unsuitable treatment processes for the incoming wastewater, yet the associated wastewater volumes are still reported as treated (Qadir et al., 2010). Therefore, it is possible that the actual treated volume of wastewater is somewhat below our estimated 52% and the proportion of collected wastewater which is not treated could far exceed 16%. 'Wastewater treatment' is also a generic term that may refer to any form of wastewater treatment regardless of level (e.g. primary, secondary or tertiary), which this study does not attempt to distinguish between. This is due to different data sources reporting different levels of treatment, for instance with GWI only reporting secondary treatment or above whilst FAO-AQUASTAT also includes primary treatment. In percentage terms, wastewater treatment by economic classification is broadly in line with previous work (Sato et al., 2013), who estimate



wastewater treatment to be 70%, 38%, 28% and 8% for high income, upper middle income, lower middle income and low income countries, respectively, compared to our quantifications of 74%, 43%, 26% and 4.2%. Whilst similar, these estimations could potentially indicate that percentage collection and treatment have increased in the developed world, but have decreased in the developing world. This could be caused by wastewater production, particularly in the developing world, rising at a faster

pace than the development of collection infrastructure and treatment facilities (Sato et al., 2013). The drivers behind wastewater re-use are a complex mixture of social, economic, geographic and hydrological drivers, and data is highly limited globally. Nevertheless, this study represents the first attempt to quantify intentionally treated wastewater re-use at the country scale. It should be noted that this study does not aim to quantify either de-facto (unintentional) treated wastewater re-use or any form (intentional or unintentional) untreated wastewater re-use. The total volume of wastewater re-used for human purposes is

therefore likely much greater than the 40.7 billion $m^3 yr^{-1}$ of intentional treated wastewater re-use estimated in this study. For example, previous research has indicated that the magnitude of intentional untreated wastewater re-use may be approximately ten times greater than intentional treated wastewater re-use (Scott et al., 2010).

This study sought to downscale country level wastewater estimates to spatially-explicit (grid-based) quantifications for

purposes such as large-scale water resource assessments and water quality modelling. Wastewater production has previously been quantified based only on simulated return flows in hydrological models (Flörke et al., 2013). Whilst our results also rely on this approach, we instead used the proportions of return flows to downscale our country-based volumes of wastewater production. Our results also represent the first efforts to quantify global wastewater collection, treatment and re-use at the sub-national level. Our validation results suggest that our downscaled estimates of wastewater treatment and re-use are, in general,

realistic. However, a number of uncertainties should also be considered. Firstly, our downscaling for wastewater production inherently relies on the accuracy of PCR-GLOBWB 2 to simulate domestic and industrial return flows proportionally. The accuracy of downscaled wastewater collection and treatment relies on the assumption this preferentially occurs in areas where wastewater production is highest. Due to the high capital costs of wastewater treatment plants, combined with economies of scale, we deem this a logical assumption (Hernández-Chover et al., 2018; Hernandez-Sancho et al., 2011). Wastewater re-use

is downscaled with the only additional criteria being a water scarcity threshold. Whilst water scarcity is an important driver of wastewater re-use, site-specific social, economic and political factors will also have a large influence on the viability of wastewater re-use on a case-by-case basis (WWAP, 2017). Accounting for these factors is outside the scope of this study. Furthermore, uncertainties in the validation datasets, both in terms of treatment capacity and geographical location, must also be recognised. Overall, due to the global scale of this work and the available data for validation, we purposely opt for more

simple and parsimonious approaches where possible.

This study did not target acreage in its considerations of wastewater re-use, which has been a common method in previous work. For example, estimates made a decade ago suggest that up to 200 million farmers practice wastewater irrigation over an area of 4.5 – 20.0 million ha worldwide (Jiménez and Asano, 2008; Raschid-Sally and Jayakody, 2008). More recently, a



global, spatially explicit assessment of irrigated croplands influenced by municipal wastewater flows estimated the area under direct and indirect wastewater irrigation at 36 million ha, of which 29 million ha are likely exposed to untreated wastewater flows (Thebo et al., 2017). These estimates were based on modelling studies and considered wastewater in both diluted and undiluted forms with a cropping intensity of 1.48 (Thebo et al., 2014). Considering the same cropping intensity and recent estimates of wastewater production (380 billion $m^3$ $yr^{-1}$, the irrigation potential of undiluted wastewater was estimated at 42

million ha (Qadir et al., 2020).

Our results have a range of important applications including as input data for water resource assessments and for as a baseline for informing and evaluating economic and management policies related to wastewater. For example, our data can be used to assess progress towards SDG 6.3 aimed at halving the proportion of untreated wastewater discharged into water bodies. As

our data is standardised for 2015 and provides full geographic coverage, problems of discrepancies in data reporting years and missing data are overcome. Similarly, our data allows for identification of hotspot regions whereby the proportion of wastewater collected and treated are low, and of areas where large volumes of wastewater are entering the environment untreated (Figure 6). Volumetrically, substantial untreated wastewater flows to the environment are found particularly across South and Southeast Asia, particularly in the populous regions of Pakistan, Malaysia, Indonesia, India and China. Information

on untreated wastewater flows have a diverse range of important implications for global water quality modelling and human health assessments.

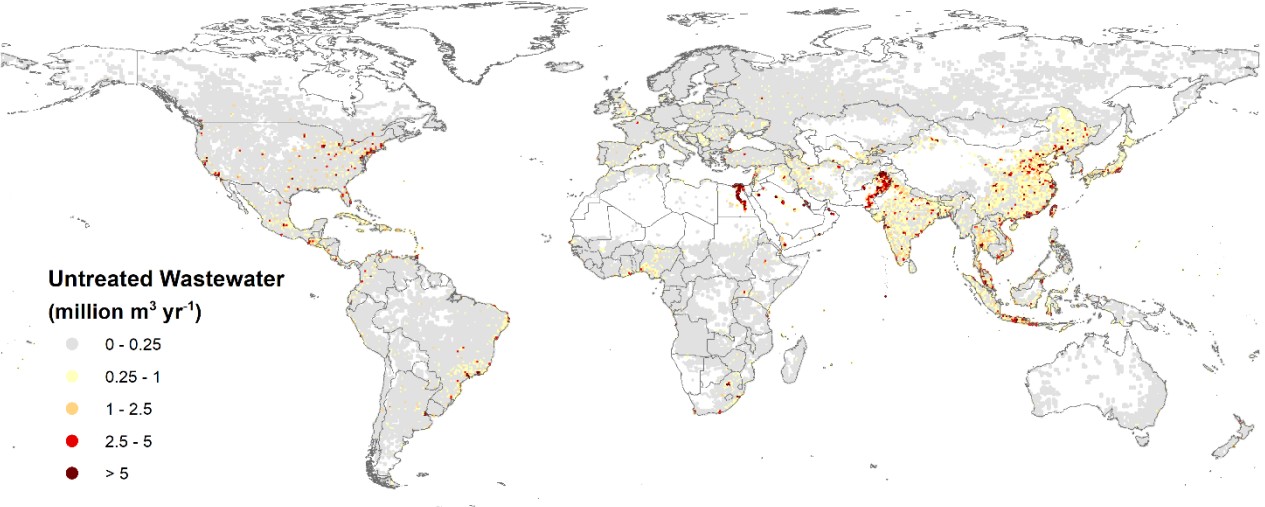

**Figure 6.** Gridded untreated wastewater flows to the environment (million $m^3$ $yr^{-1}$) at 5 arc-minute spatial resolution.

Our results also highlight the vast potential of treated wastewater as an unconventional water resource for augmenting water resources and alleviating water scarcity, particularly in water scarce regions. To put wastewater as a potential resource into perspective, its estimated global volume of ~360 $km^3$ $year^{-1}$ is comparable to the global consumptive use of non-renewable



groundwater of 150-400 km$^3$ year$^{-1}$ over the years 2000-2010 (Bierkens and Wada, 2019). As wastewater production continues to rise with population and economic growth, wastewater management and re-use practices will become more important in the future (WWAP, 2017). Expansion in re-use of wastewater must be accompanied by strong legislation and regulations to ensure its safety (Smol et al., 2020; Voulvoulis, 2018). However, in response to concerns related to groundwater contamination, disruption to industrial processes and impacts for human health, tightening regulation can also be a barrier to expansion in treated wastewater re-use (Voulvoulis, 2018). It should also be recognised that wastewater re-use is not viable in all regions due to economic, technical and social considerations (Voulvoulis, 2018). Particularly in water-scarce developing nations with economic constraints, the application of untreated wastewater (diluted or undiluted) will likely remain the dominant form of wastewater re-use (Qadir et al., 2010). This is especially true in dry areas, despite official restrictions and regardless of potential health implications, where untreated wastewater re-use is triggered because (1) wastewater is a reliable or often the only guaranteed water source available throughout the year; (2) the need to apply fertilisers decreases as wastewater is a source of nutrients; (3) wastewater re-use can be cheaper and less energy intensive than other water sources, such as if the alternative clean water source is deep groundwater; and (4) additional economic benefits such as higher income generation from the cultivation and marketing of high-value crops such as vegetables, creating year-round employment opportunities.

Continued failure to address wastewater as a major social and environmental challenge prohibits progress towards the 2030 Agenda for Sustainable Development (WWAP, 2017). Ultimately, the cost of action must also be weighed against the cost of inaction (Hernández-Sancho et al., 2015). A paradigm shift in wastewater management is required from viewing wastewater as solely an environmental problem associated with pollution control and regulations, to recognise the economic opportunities of wastewater, which can provide a means of financing management and treatment (Wichelns et al., 2015; WWAP, 2017). In addition to revenue from selling treated wastewater for re-use, these opportunities include 'fit-for-purpose' treatment (Chhipi-Shrestha et al., 2017), recovery of energy and nutrients (Qadir et al., 2020) and cascading re-use of water from high to lower quality (Hansen et al., 2016). Creative exploitation of these opportunities offers potential to support the transition to a circular economy (Smol et al., 2020; Voulvoulis, 2018) and make progress towards many interconnected SDGs such as achieving a water-secure future for all (WWAP, 2017).

**Author contribution**

ERJ performed the analyses, drafted the manuscript and developed the study with input of MTHvV and MFPB. MTHvV, MQ and MFPB provided feedback and guidance throughout the entire process. All authors contributed to and approved the manuscript.

**Competing interests**

The authors declare no competing financial interests.





**Acknowledgements**

The authors are grateful to Dr. Edwin Sutanudjaja for providing output data from PCR-GLOBWB2 to support this work. MQ appreciates support of the Government of Canada for UNU-INWEH through Global Affairs Canada.



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
