# Peer review of "Country-level and gridded estimates of wastewater production, collection, treatment and re-use."

_Earth System Science Data, 2020_

## Referee Comment (RC1) · Anonymous Referee #1 · 30 Oct 2020

The paper by Jones et al. provides a revised and consistent global outlook of the state of wastewater production, collection, treatment and re-use. It uses available country-level wastewater data and regression analyses to estimate information where it is unavailable. The year selected for the country-level data was 2015, and unavailable data were standardized to the same year using relationships with GDP. In addition, the authors downscaled the country-level data to a 5-min resolution grid using return flow data from the global water balance model PCR-GLOBWB. The downscaling was validated using European, US, and some global (yet less in numbers) records of wastewater treatment plants. Validation efforts delivered reasonable model performance indicators, and uncertainties were estimated using a bootstrapping technique.

[Figure]

The final data product provides a gridded map that includes quantities of wastewater production, collection, treatment and re-use at 5-min spatial resolution. Global water quality models and large-scale water assessments have been lacking this type of information in the past, so this paper is clearly a very important addition to the field. As this is a global effort, there are severe constraints regarding data availability and quality. The authors developed (and explained) reasonable approaches to overcome these problems. While some of their methods are based on speculation regarding the relevant processes, the ultimate test of such an approach is the validation of the results. I think the authors did a commendable job in their validation and comparisons, and I do appreciate that they reveal important shortcomings and clearly state that the results must be interpreted with caution.

So overall I think this paper presents an excellent global-scale effort to generate an advanced and novel gridded map of wastewater quantities. The manuscript is generally well written and very clearly structured. I strongly recommend its publication. I have a series of mostly minor comments that I list below. They are all written with the intention to further improve the manuscript. I also want to express my thanks to the authors for providing this important dataset to the research community!

Individual comments (line numbers refer to PDF version):

1. Title (and elsewhere): After reading the title ("Spatially-explicit ...") my initial expectation was that the paper will describe explicit locations of wastewater production and collection (e.g. locations of wastewater treatment plants). Only when reading the manuscript, I realized that the dataset refers to a modeled distribution of wastewater quantities at sub-national scale. This is still great, but maybe a slightly different title could help avoiding this confusion, such as "Downscaled gridded model estimates of global wastewater..." or you could at least refer to "model estimates" rather than just "estimates".

2. There is no distinction made in the dataset between industrial and domestic wastew-

ater - which have very different characteristics and effects on environmental waters. It would thus be great to briefly discuss whether and how the combination of domestic and industrial wastewater may cause problems in the dataset, and in particular in the downscaling process. For example, industrial wastewater can be produced at locations with little correlation to population centers, i.e. at very distinct or remote locations compared to domestic wastewater - is this accounted for in the modeled return flows of PCR-GLOBWB? Also, I assume that the validation data cannot clearly distinguish between domestic and industrial wastewater as well? Related to this: Line 58 states for the first time that throughout the manuscript, domestic and industrial wastewater are not considered separately but lumped. As this is very important, it could be emphasized more, e.g. by referring more clearly to "combined domestic and industrial sources". This could also be done at other locations, where appropriate.

3. Lines 62 and following: You state that "wastewater treatment improves the quality of 'used' water resources" and this notion seems to prevail throughout the introduction and discussion. But while "wastewater treatment" as a process certainly has the GOAL to improve water quality, what about the fact that substances that are not or cannot be treated by treatment facilities can cause the opposite effect: in these cases, wastewater treatment plants can represent point sources of pollution, especially in the case of emerging contaminants. This has not been addressed in the paper and I thus encourage the authors to at least briefly reflect on the issue.

4. Table 1 (Standardisation to 2015): There is very little explanation in the text about the rationale behind the standardization methods. It seems the main assumption is a linear behavior of wastewater amounts based on GDP, right? This could briefly be mentioned in the text. Also, for collection and treatment: I do not understand why the values are divided by GDP per capita but then not multiplied by GDP per capita but by total GDP. Is this just a typo?

5. Figure 3: Very interesting figure. The one country that stands out to me as a surprise in wastewater production is Egypt. The Nile and Nile delta show also exceptionally high

values in Figures 4 and 6. There is no comment about Egypt in the manuscript. Any explanations on why Egypt has so high domestic and industrial wastewater amounts?

6. To my knowledge, the European dataset of wastewater treatment plants reports treatment capacity only in Population Equivalent (PE); however, the manuscript states that the volume flow rate was obtained based on a linear regression for plants reporting both parameters (PE and volume). It would be interesting to know how many plants included this information since the openly available dataset at the EEA website seems not to provide the volume flow rate.

7. Line 437 and following: You say: "This may occur due to discrepancies between the design (i.e. maximum) capacity of wastewater treatment plants . . ." Ok, that is a fair point. But the other option is that your model overpredicts in places without treatment plants and therefore underpredicts in places with treatment plants. This potential model bias should be acknowledged and briefly discussed.

8. Line 244-245: "For validating downscaled wastewater re-use, only plants (with treatment capacity > 1 million m3 yr-1) using tertiary or higher wastewater treatment technologies were considered." The rationale for this decision is not clear to me.

9. Figure 1 shows that there are data for only $\sim$half of global countries available (e.g. production data are available for 118 countries), yet 90% of population is covered by data. The only explanation for this is that all the high-population countries are included in those countries for which data exist, correct? It may be worth pointing this out as initially I was confused on how these numbers match up. I found some explanations later in the manuscript, but maybe this general fact could be stated earlier.

Minor comments (line numbers refer to PDF version):

Throughout the text, the authors use the expression "data" in singular form ("data is . . ."). I am more used to data in plural form ("data are . . .").

Also throughout the text, the authors use the expression "whilst" (many times). It

is my understanding that "whilst" may be perceived as 'archaic' in American English (https://en.wikipedia.org/wiki/While). So maybe use "while" instead?

Line 24 (and possibly elsewhere): The expression "significant" is often reserved for instances where it refers to statistical methods. Here, an alternative might be to use "substantial".

Line 25: replace "containing" with "comprising"?

Line 28: I suggest spelling out the first occurrence of SDG here (or remove the example)

Line 80: "that ensure" instead of "to ensure"?

Line 88: rephrase "whereby ..." - maybe "which includes a target that ..."

Line 89: add a space in "SDG 6.3"

Line 98: say "to a grid level of 5 arc-minute spatial resolution"

Figure 1: I cannot find the letters (a), (b) and (c) reflected in the figure, so it is difficult to find out which panels the caption refers to. Also, I suggest removing the asterisk from the figure and simply add the definition of 'population coverage' as part of the normal caption.

Line 117: add "(Table 1)" after "... databases"

Lines 125-126 (and elsewhere): I find the use the acronyms, the article "the", as well as the verbs not consistent here. I would say "GWI reports ... whereas FAO reports ... and UNSD reports" etc.

Line 126: say "... reported by UNSD"

Lines 139-140: repetitive use of "both" - delete one?

Table 1: start title with "Wastewater data sources and population coverage by region and economic aspects..." Also, the footnotes of the table could be shortened. E.g. the

square brackets always refer to the number of countries, so this could be explained once in the title and does not need to be repeated for each of the footnotes.

Line 166 (and possibly elsewhere): there are some instances where "per capita" is written as "per Capita" (capitalized)

Line 168-169: "Data was transformed, as appropriate, to ensure normality." This statement is not clear to me. Does it refer to what is called "sqrt" in Table 3, which I guess means that the square-root of values was calculated? Could both instances be clarified, e.g. by adding a little more information in this sentence here?

Table 2: spelling of "Agricultural Land" should be "Agricultural land"

Line 201: The eight regions are listed here for the first time, but no explanation is provided on what exactly defines these regions. Is this some official breakdown so that one could look up the countries that belong to each region?

Line 215: "occur" instead of "occurs"

Line 241: repetition of "both"

Line 243: "where aggregated PER CELL"?

Line 248: "were" instead of "was"

Line 254 and 257: While most explanations in this paragraph are in the past tense, two verbs are in present tense ("is expanded" and "is then compared"). Typo?

Line 261: "were" instead of "was"

Line 289: "Human Development Index" (as it was also capitalized elsewhere)

Line 290-291: say "was found to have the strongest influence on"

Line 311: "nations" (plural)

Figure 2: Could add the number of countries that are displayed in each panel, e.g. add

"n = ...". Also, the graphs may be clearer if using white instead of grey background.

Line 326: move "billion" after parentheses

Line 331: "being from" instead of "from"

Line 353: "World Health Organization's" (with apostrophe)

Line 362: "indicate" instead of "indicates"

Line 379: add commas before and after "respectively"

Line 391: delete "regions"

Line 397: "treat" instead of "treated"

Line 398: "than in" instead of "than by"

Table 4, title: Start with "Wastewater production, collection, treatment and re-use (billion m3 yr-1) by region and economic development level." I suggest changing "brackets" to "parentheses" (as brackets in American English would refer to squared brackets). And say "regressions" (plural).

Line 421: "yr-1" instead of "/year". Also, "occur" instead of "occurs"

Line 422: "collection . . . and treatment . . . are" (instead of "is")

Line 426: "with only available wastewater resources"—not clear, wrong wording?

Line 456: I suggest using "underpinning source data" instead of "underlying source data"

Line 472: "upon which country-level estimates incorporate" seems to be incorrect wording

Line 479: I suggest breaking this very long paragraph into 2 here

Line 486: say "factors" instead of "drivers" to avoid repetition

Line 489: say "of untreated" (add "of")

Line 496: "Whilst our results also rely on this approach, we instead used..." This sounds odd (first you say you do the same, then you say 'instead') - rephrase?

Line 512: The expression "acreage" may not be known to all readers. Could just say "spatial extent"

Line 519: close the parentheses

Line 522: "for as a baseline for"? should this be "as a baseline for"?

Line 525: "... problems of discrepancies in data reporting years and missing data are overcome." It sounds quite optimistic that the problems are truly "overcome" (i.e. solved). Maybe say "reduced" instead?

Lines 528-529: repetitive use of "particularly"

Lines 549-551: you use the word "such" three times here

Lines 550-551: The grammar/verb of the description of point (4) seems incorrect. Change "creating" to "create"?

---

## Referee Comment (RC2) · Anonymous Referee #2 · 17 Nov 2020

This study provided the comprehensive and consistent global outlook on the state of wastewater production collection treatment and reuse. And the country level wastewater data are downscaled and validated at 5 arc- minute resolution. Its results represent the first efforts to global wastewater collection treatment and reuse at the subnational level. It is a very interesting and useful work for the wastewater research. And the quality of the data set as submitted is high. The data analysis and discussions are sufficient. So I think it prepared well for publication. It analyzed the relationship among the production, collection, treatment and reuse of wastewater, the income level and the population. However, I think the influence from the pollution of agriculture, especially for the global grain production areas, can not be ignored. So I suggest the author to

add the the analysis or discussion of this part. It may be more perfect.

---

## Author Comment (AC1) · 27 Nov 2020

**Response to comments of reviewer 1**

| | |
|---|---|
| Manuscript: | essd-2020-156 |
| Original title: | Spatially-explicit estimates of global wastewater production, collection, treatment and re-use. |
| Revised title: | Country-level and gridded estimates of wastewater production, collected, treatment and re-use. |
| Authors: | Edward R. Jones, Michelle T.H. van Vliet, Manzoor Qadir, Marc F. P. Bierkens |

**General comments**

The paper by Jones et al. provides a revised and consistent global outlook of the state of wastewater production, collection, treatment and re-use. It uses available country-level wastewater data and regression analyses to estimate information where it is unavailable. The year selected for the country-level data was 2015, and unavailable data were standardized to the same year using relationships with GDP. In addition, the authors downscaled the country-level data to a 5-min resolution grid using return flow data from the global water balance model PCR-GLOBWB. The downscaling was validated using European, US, and some global (yet less in numbers) records of wastewater treatment plants. Validation efforts delivered reasonable model performance indicators, and uncertainties were estimated using a bootstrapping technique.

The final data product provides a gridded map that includes quantities of wastewater production, collection, treatment and re-use at 5-min spatial resolution. Global water quality models and large-scale water assessments have been lacking this type of information in the past, so this paper is clearly a very important addition to the field. As this is a global effort, there are severe constraints regarding data availability and quality. The authors developed (and explained) reasonable approaches to overcome these problems. While some of their methods are based on speculation regarding the relevant processes, the ultimate test of such an approach is the validation of the results. I think the authors did a commendable job in their validation and comparisons, and I do appreciate that they reveal important shortcomings and clearly state that the results must be interpreted with caution. So overall I think this paper presents an excellent global-scale effort to generate an advanced and novel gridded map of wastewater quantities. The manuscript is generally well written and very clearly structured. I strongly recommend its publication. I have a series of mostly minor comments that I list below. They are all written with the intention to further improve the manuscript. I also want to express my thanks to the authors for providing this important dataset to the research community!

We thank the reviewer very much for the insightful comments and suggestions. We are pleased to read that the reviewer is complimentary about the manuscript and recognises the value of our dataset for the research community. The reviewer addresses a number of important topics, which we overall agree with and which have all been addressed in the revised manuscript. Please see our point-by-point responses to the individual comments below.

**Individual comments**

1. Title (and elsewhere): After reading the title ("Spatially-explicit . . .") my initial expectation was that the paper will describe explicit locations of wastewater production and collection (e.g. locations of wastewater treatment plants). Only when reading the manuscript, I realized that the dataset refers to a modeled distribution of wastewater quantities at sub-national scale. This is still great, but maybe a slightly different title could help avoiding this confusion, such as "Downscaled gridded model estimates of global wastewater. . ." or you could at least refer to "model estimates" rather than just "estimates".

We understand this source of confusion for the reviewer. We propose to change the provisional title to: "*Country-level and gridded estimates of wastewater production, collection, treatment and re-use*". We have chosen not to include the word 'modelled' in our dataset, as we do not want to give off the impression that the results are exclusively modelled (as the study is primarily underpinned by reported data at the country-level).

2. There is no distinction made in the dataset between industrial and domestic wastewater - which have very different characteristics and effects on environmental waters. It would thus be great to briefly discuss whether and how the combination of domestic and industrial wastewater may cause problems in the dataset, and in particular in the downscaling process. For example, industrial wastewater can be produced at locations with little correlation to population centers, i.e. at very distinct or remote locations compared to domestic wastewater - is this accounted for in the modeled return flows of PCR-GLOBWB? Also, I assume that the validation data cannot clearly distinguish between domestic and industrial wastewater as well? Related to this: Line 58 states for the first time that throughout the manuscript, domestic and industrial wastewater are not considered separately but lumped. As this is very important, it could be emphasized more, e.g. by referring more clearly to "combined domestic and industrial sources". This could also be done at other locations, where appropriate.

PCR-GLOBWB calculates water use (i.e. withdrawals, consumption and hence return flows) from the domestic and industrial sectors individually, which we have then lumped together for the downscaling procedure. Since the initial manuscript submission, we have further updated the return flows used for the downscaling procedure using a more recent water use dataset, developed at 5arc-min, from Water Futures and Solutions (WFaS) initiative (Wada et al., 2016). The use of this more recent dataset also facilitated an improved downscaling methodology and results.

In both the WFaS and PCR-GLOBWB methodologies, domestic demands are calculated on the basis of population and a country-specific per capita water use. Conversely, industrial demands are calculated on the basis of four socio-economic variables (GDP, electricity production, energy consumption and household consumption) (see Wada et al., 2011; Wada et al., 2014 and Wada et al., 2016 for details). Thus, industrial demand (and hence return flows) are simulated in areas taking into account multiple variables aside from just population. Regarding the question of the reviewer, the industrial return flows used do indeed account for industrial flows also being located in areas outside population centers.

We agree that the composition of industrial and domestic wastewater is different and that both have different environmental effects. We also strongly agree that a distinction between industrial and domestic wastewater is important for different applications. However, we choose to lump these flows for now. Domestic and industrial return flows are typically collected in the same (municipal) sewers before conveyance to wastewater treatment. As the reviewer points out, a distinction between domestic and industrial wastewater treatment cannot be validated individually for this reason. We

therefore prefer to present aggregated results of municipal wastewater, including both domestic and industrial wastewater.

The manuscript will be been updated as appropriate to reflect these changes in using this more recent WFaS water use dataset (Wada et al, 2016).

3. Lines 62 and following: You state that "wastewater treatment improves the quality of 'used' water resources" and this notion seems to prevail throughout the introduction and discussion. But while "wastewater treatment" as a process certainly has the GOAL to improve water quality, what about the fact that substances that are not or cannot be treated by treatment facilities can cause the opposite effect: in these cases, wastewater treatment plants can represent point sources of pollution, especially in the case of emerging contaminants. This has not been addressed in the paper and I thus encourage the authors to at least briefly reflect on the issue.

This is an excellent point and we entirely agree that this should be considered in the manuscript. We recognise that wastewater collection and treatment processes, if insufficient, can concentrate particular pollutants (especially for emerging pollutants) and thus represent a point source for environmental contamination. We propose to add some sentences in the discussion section to reflect this:

*"It should be noted that while the aim of wastewater collection and treatment is to reduce pollutant loadings to minimise risks to human health and the environment, these facilities can also act as point sources of pollution. Wastewater collection concentrates pollutants which, can pose serious water quality issues if discharged with insufficient treatment. Furthermore, a range of emerging pollutants (e.g. pharmaceuticals, pesticides and industrial chemicals) are concentrated in wastewater collection networks (Geissen et al., 2015). These pollutants are of particular concern as they are not typically monitored for or sufficiently removed in wastewater treatment processes, with ambiguous risks posed to human and environmental health even in low concentrations (Deblonde et al., 2011; Geissen et al., 2015). The solution is not however to collect less wastewater, but to increase treatment in terms of percentage of collected wastewater, treatment level and the number of pollutants (UNEP, 2016)."*

4. Table 1 (Standardisation to 2015): There is very little explanation in the text about the rationale behind the standardization methods. It seems the main assumption is a linear behavior of wastewater amounts based on GDP, right? This could briefly be mentioned in the text. Also, for collection and treatment: I do not understand why the values are divided by GDP per capita but then not multiplied by GDP per capita but by total GDP. Is this just a typo?

For the standardisation to 2015, we indeed assumed a linear behavior with on GDP (for wastewater production) and GDP per capita (for wastewater collection and treatment). We propose to add a line to clarify and justify this choice in the text:

"*Wastewater production is assumed to be dependent upon both population size and per capita production (related to per capita wealth). Hence, we standardise wastewater production linearly with GDP, a combined metric of population size and wealth. Conversely, wastewater collection and treatment are assumed to be more dependent on economics, hence we linearly apply GDP per capita for standardisation*".

We also would like to thank the reviewer for bringing the issue with GDP vs. GDP per capita to our attention. This is indeed an unfortunate typo. We will make the corrections to Table 1 and confirm that GDP per capita is indeed the correct variable that was used for standarising collection and treatment.

5. Figure 3: Very interesting figure. The one country that stands out to me as a surprise in wastewater production is Egypt. The Nile and Nile delta show also exceptionally high values in Figures 4 and 6. There is no comment about Egypt in the manuscript. Any explanations on why Egypt has so high domestic and industrial wastewater amounts?

We agree that Egypt is a particularly interesting country, and stands out in Figures 3, 4 and 6.

It is worth noting that wastewater data for Egypt was cross referenced from three sources: 1) GWI 2015: 13,623 million $m^3$ $yr^{-1}$; 2) UNSD 2015: 11,899 million $m^3$ $yr^{-1}$; and 3) Aquastat 2012: 6,497 million $m^3$ $yr^{-1}$ which, standarised to 2015, was 8,429 million $m^3$ $yr^{-1}$. Whilst some variation exists between these numbers, data from all three sources indicate a relatively high per capita wastewater production (between 91 – 147 $m^3$ $yr^{-1}$ per capita). The final value determined (mean average from the three sources for 2015 data) for Egypt as 11,317 million $m^3$ $yr^{-1}$ (122 $m^3$ $yr^{-1}$ per capita) – which is relatively high (compared, for instance, to regional averages). These high values are further corroborated by estimates of Egypt's domestic water use (200 l per capita per day in 2007), almost double that of Germanys (data from National Water Research Centre (Egypt). Next to this, we expect that the domestic + industrial return flows (or wastewater) values may also be high due to the relatively low water use efficiencies and system losses.

In terms of the downscaled maps (Figures 4 and 6), population density is an important variable for quantification of domestic and industrial return flows in PCR-GLOBWB. Egypt's population is of course heavily concentrated along the banks of the Nile and in the Nile Delta. This heavily concentrates the downscaled wastewater into relatively few gridcells (relative to the size of Egypt).

6. To my knowledge, the European dataset of wastewater treatment plants reports treatment capacity only in Population Equivalent (PE); however, the manuscript states that the volume flow rate was obtained based on a linear regression for plants reporting both parameters (PE and volume). It would be interesting to know how many plants included this information since the openly available dataset at the EEA website seems not to provide the volume flow rate.

The reviewer makes a good point here with regards to our need to convert wastewater treatment in population equivalent to volume flow rate for validation purposes. We indeed used a linear relation between Population Equivalent (PE) and volume flow rate for wastewater treatment plants that reported both of these variables, and applied this to plants where only the PE was reported.

Fortunately, we were able to obtain a decent number of data points both for the global database (GWI) and specifically for Europe (EEA) in order to do the linear regression.

For the EEA dataset, both PE and volume flow rate (in $m^3$ $yr^{-1}$) data were available for 8,309 treatment plants. The linear relationship was applied to 17,593 plants with only treatment capacity in PE, to estimate treatment capacity in volume flow rate units. Predicted vs. reported wastewater treatment capacity is displayed for the EEA plants where both PE and volume flow rate units (e.g. $m^3$ $yr^{-1}$) are reported in Figure R1 ($R^2 = 0.80$).

For the GWI dataset, this was available for 227 wastewater treatment plants, with the linear relation applied to 83 plants with capacity only reported in PE. The remaining plants has their capacities only reported in volume flow units. It is important to note that wastewater treatment plants located in the US and Europe were excluded from the GWI dataset to avoid plants potentially being counted multiple times if contained in the GWI dataset + the EEA or US EPA dataset. Predicted vs. reported wastewater treatment capacity is displayed for the GWI plants where both PE and volume flow rate units (e.g. $m^3$ $yr^{-1}$) are reported in Figure R2 ($R^2 = 0.81$).

All wastewater data from the US EPA were reported in volume flow rate units (million gallons per day) which was converted into $m^3$ $yr^{-1}$.

With regards to the EEA data, we found this data to be freely available. If the reviewer is interested in this dataset, they can download this information from Waterbase (https://www.eea.europa.eu/data-and-maps/data/waterbase-uwwtd-urban-waste-water-treatment-directive-6) and consult the worksheet named "UWWTPS". Providing this dataset has not since been adjusted, information for plant latitude (column AF); longitude (column AH); load entering UWWTP in PE (column AG) and wastewater treated (column BP) in $m^3$ $yr^{-1}$ can be obtained. Alternatively, the reviewer is welcome to contact us directly if they would like the plant level data for the EU. As part of this paper, we have already collated important information for them various spreadsheets provided by the EEA into a single spreadsheet and undertaking some basic formatting.

[Figure]

**Figure R1.** Predicted vs. reported wastewater treatment for EU wastewater treatment plants reporting treatment capacity in both population Equivalent (PE) and in volume flow rate units (e.g. million $m^3$ $yr^{-1}$).

[Figure]

**Figure R2.** Predicted vs. reported wastewater treatment for GWI wastewater treatment plants reporting treatment capacity in both population Equivalent (PE) and in volume flow rate units (e.g. million $m^3$ $yr^{-1}$).

7. Line 437 and following: You say: "This may occur due to discrepancies between the design (i.e. maximum) capacity of wastewater treatment plants . . ." Ok, that is a fair point. But the other option is that your model overpredicts in places without treatment plants and therefore underpredicts in places with treatment plants. This potential model bias should be acknowledged and briefly discussed.

We agree that this potential bias might also explain these discrepancies. We think this is ultimately depends on the downscaling method for wastewater production (i.e. using modelled return flow data from PCR-GLOBWB), on which quantifications of wastewater treatment are based. We propose to explain this in more detail by adding to the text:

*Furthermore, uncertainties in the data used as basis for downscaling of wastewater production (i.e. PCR-GLOBWB return flows) directly impacts the downscaled results of wastewater treatment. For example, the underprediction of return flows in urban areas and overprediction in rural areas could lead to the overprediction of wastewater treatment in areas without treatment plants and underprediction of wastewater treatment for grid cells with large treatment capacities.*

8. Line 244-245: "For validating downscaled wastewater re-use, only plants (with treatment capacity > 1 million m3 yr-1) using tertiary or higher wastewater treatment technologies were considered." The rationale for this decision is not clear to me.

Unfortunately, plant-level data on wastewater re-use is limited. Due to the lack of data, assumptions had to be made for validating wastewater re-use. Aside for the limited number of plants specifically designated as re-use facilities from GWI (78 wastewater re-use plants), we assumed that tertiary or

higher wastewater treatment processes likely are most strongly related to wastewater re-use (as these treatment levels are a pre-requisite for safe re-use for most applications). We narrowed the scope to larger wastewater treatment plants only (>1 million m³ yr⁻¹), where tertiary+ treated wastewater is potentially available for re-use in substantial volumes and to exclude highly localised wastewater treatment facilities (e.g. for specific industrial facilities) which are very difficult to capture in lieu of poor data availability.

9. Figure 1 shows that there are data for only ∼half of global countries available (e.g. production data are available for 118 countries), yet 90% of population is covered by data. The only explanation for this is that all the high-population countries are included in those countries for which data exist, correct? It may be worth pointing this out as initially I was confused on how these numbers match up. I found some explanations later in the manuscript, but maybe this general fact could be stated earlier.

The reviewer is correct. Reported data was typically more available for large countries as opposed to smaller ones. We propose to make this more clear in the text when the comparison between % population and number of countries with reported data is first mentioned:

*"Reported wastewater data was available for the majority of the world's most populous countries. This results in a the high percentage population coverage relative to the number of countries."*

**Minor Comments**

Throughout the text, the authors use the expression "data" in singular form ("data is . . ."). I am more used to data in plural form ("data are . . .").

We agree with the reviewer. This will be corrected throughout the manuscript.

Also throughout the text, the authors use the expression "whilst" (many times). Its my understanding that "whilst" may be perceived as 'archaic' in American English (https://en.wikipedia.org/wiki/While). So maybe use "while" instead?

We will change all occurrences of "whilst" to "while", the more familiar formulation of the word in American English.

Line 24 (and possibly elsewhere): The expression "significant" is often reserved for instances where it refers to statistical methods. Here, an alternative might be to use "substantial".

We thank the reviewer for this comment, and agree that "substantial" is a better word choice for instances not referring to statistical methods. This will be adjusted throughout the manuscript.

Line 25: replace "containing" with "comprising"?

This will been changed.

Line 28: I suggest spelling out the first occurrence of SDG here (or remove the example)

We propose to remove the example of the SDGs in the abstract for readability. The anacronym SDG will now spelt out in the first occurrence in the manuscript instead.

Line 80: "that ensure" instead of "to ensure"?

This will been changed.

Line 88: rephrase "whereby . . ." - maybe "which includes a target that . . ."

This line will be reformulated in line with the recommendation to rephrase.

Line 89: add a space in "SDG 6.3"

This will be changed.

Line 98: say "to a grid level of 5 arc-minute spatial resolution"

This will be changed.

Figure 1: I cannot find the letters (a), (b) and (c) reflected in the figure, so it is difficult to find out which panels the caption refers to. Also, I suggest removing the asterisk from the figure and simply add the definition of 'population coverage' as part of the normal caption.

Thanks for noticing that (a), (b) and (c) was missing from the figure. This was a mistake and will be corrected. The asterisk will be removed as recommended, and the figure caption has been updated. Figure panels b and c will also be increased slightly in size for better readability.

Line 117: add "(Table 1)" after ". . . databases"

This will be added.

Lines 125-126 (and elsewhere): I find the use the acronyms, the article "the", as well as the verbs not consistent here. I would say "GWI reports . . . whereas FAO reports . . . and UNSD reports" etc.

This will be changed in the suggested way.

Line 126: say ". . . reported by UNSD"

This will be changed.

Lines 139-140: repetitive use of "both" - delete one?

This will be changed.

Table 1: start title with "Wastewater data sources and population coverage by region and economic aspects. . ." Also, the footnotes of the table could be shortened. E.g. the square brackets always refer to the number of countries, so this could be explained once in the title and does not need to be repeated for each of the footnotes.

Thanks for this suggestion to alter the title and footnotes. These will be changed.

Line 166 (and possibly elsewhere): there are some instances where "per capita" is written as "per Capita" (capitalized)

"per capita" (lowercase) will be used consistently throughout the manuscript.

Line 168-169: "Data was transformed, as appropriate, to ensure normality." This statement is not clear to me. Does it refer to what is called "sqrt" in Table 3, which I guess means that the square-root of values was calculated? Could both instances be clarified, e.g. by adding a little more information in this sentence here?

Some variables (e.g. GDP, population) were log-transformed to limit skew in the independent variables (which can vary across many orders of magnitude) and to achieve normality. Alternatively, square-root (sqrt) transformation is used for desalination capacity per capita as this data-set contained zero values (countries with no desalination) and thus is inappropriate for log transformation. We propose to further clarified this in the text:

*"Data was transformed, either using a log or square root transformation, to reduce the skew in the independent variables and to ensure normality."*

Table 2: spelling of "Agricultural Land" should be "Agricultural land"

The appropriate change will be made.

Line 201: The eight regions are listed here for the first time, but no explanation is provided on what exactly defines these regions. Is this some official breakdown so that one could look up the countries that belong to each region?

Thanks for this. The regional classifications are taken from the World Bank, the same as for the countries and income classifications. We agree this should be more clear in the text, so we will add this. This information is also contained in the dataset associated with the publication.

Line 215: "occur" instead of "occurs"

This will been changed.

Line 241: repetition of "both"

One occurrence of the word "both" will be removed.

Line 243: "where aggregated PER CELL"?

Correct. This will be added.

Line 248: "were" instead of "was"

This will be changed.

Line 254 and 257: While most explanations in this paragraph are in the past tense, two verbs are in present tense ("is expanded" and "is then compared"). Typo?

Thanks for noticing this. This will be changed.

Line 261: "were" instead of "was"

This will be changed.

Line 289: "Human Development Index" (as it was also capitalized elsewhere)

This will be changed.

Line 290-291: say "was found to have the strongest influence on"

Good suggestion. This will be changed.

Line 311: "nations" (plural)

This will be changed.

Figure 2: Could add the number of countries that are displayed in each panel, e.g. add "n = ...". Also, the graphs may be clearer if using white instead of grey background.

We will add the number of countries in each panel, as suggested. We prefer to keep the backgrounds of the graphs grey.

Line 326: move "billion" after parentheses

This will be changed.

Line 331: "being from" instead of "from"

This will be changed.

Line 353: "World Health Organization's" (with apostrophe)

This will be changed.

Line 362: "indicate" instead of "indicates"

This will be changed.

Line 379: add commas before and after "respectively"

This will be changed.

Line 391: delete "regions"

This will be changed.

Line 397: "treat" instead of "treated"

This will be changed.

Line 398: "than in" instead of "than by"

This will be changed.

Table 4, title: Start with "Wastewater production, collection, treatment and re-use (billion m3 yr-1) by region and economic development level." I suggest changing "brackets" to "parentheses" (as brackets in American English would refer to squared brackets). And say "regressions" (plural).

Thanks for this suggestion. We will alter the title as recommended.

Line 421: "yr-1" instead of "/year". Also, "occur" instead of "occurs"

This will be changed.

Line 422: "collection . . . and treatment . . . are" (instead of "is")

This will be changed.

Line 426: "with only available wastewater resources" not clear, wrong wording?

We agree this is ambiguous wording. We will make the appropriate changes in the text to clarify this.

Line 456: I suggest using "underpinning source data" instead of "underlying source data"

This will be changed.

Line 472: "upon which country-level estimates incorporate" seems to be incorrect wording

We agree this is incorrectly worded. We will change the wording in the manuscript.

Line 479: I suggest breaking this very long paragraph into 2 here

This will be changed.

Line 486: say "factors" instead of "drivers" to avoid repetition

This will be changed.

Line 489: say "of untreated" (add "of")

This will be changed.

Line 496: "Whilst our results also rely on this approach, we instead used. . ." This sounds odd (first you say you do the same, then you say 'instead') - rephrase?

We agreed that this is an 'odd' statement. We will adjust the wording to make this more clear.

Line 512: The expression "acreage" may not be known to all readers. Could just say "spatial extent"

We prefer to keep the term "acreage" here, as this is the terminology used in the referenced work.

Line 519: close the parentheses

This will be changed.

Line 522: "for as a baseline for"? should this be "as a baseline for"?

This will be changed.

Line 525: ". . . problems of discrepancies in data reporting years and missing data are overcome." It sounds quite optimistic that the problems are truly "overcome" (i.e. solved). Maybe say "reduced" instead?

Good suggestion. We will use the word "reduced" instead.

Lines 528-529: repetitive use of "particularly"

This will be changed.

Lines 549-551: you use the word "such" three times here

This will be changed.

Lines 550-551: The grammar/verb of the description of point (4) seems incorrect. Change "creating" to "create"?

This will be changed.

**References**

Deblonde, T., Cossu-Leguille, C., and Hartemann, P.: Emerging pollutants in wastewater: a review of the literature, International journal of hygiene and environmental health, 214, 442-448, 10.1016/j.ijheh.2011.08.002, 2011.

Geissen, V., Mol, H., Klumpp, E., Umlauf, G., Nadal, M., van der Ploeg, M., van de Zee, S. E. A. T. M., and Ritsema, C. J.: Emerging pollutants in the environment: A challenge for water resource management, International Soil and Water Conservation Research, 3, 57-65, https://doi.org/10.1016/j.iswcr.2015.03.002, 2015.

Wada, Y., Beek, L. P. H., Viviroli, D., Dürr, H., Weingartner, R., and Bierkens, M. F. P.: Global monthly water stress: II. Water demand and severity of water, Water Resources Research - WATER RESOUR RES, 47, 10.1029/2010WR009792, 2011.

Wada, Y., Wisser, D., and Bierkens, M. F. P.: Global modeling of withdrawal, allocation and consumptive use of surface water and groundwater resources, Earth Syst. Dynam., 5, 15-40, https://doi.org/10.5194/esd-5-15-2014, 2014.

Wada, Y., Flörke, M., Hanasaki, N., Eisner, S., Fischer, G., Tramberend, S., Satoh, Y., van Vliet, M. T. H., Yillia, P., Ringler, C., Burek, P., and Wiberg, D.: Modeling global water use for the 21st century: the Water Futures and Solutions (WFaS) initiative and its approaches, Geosci. Model Dev., 9, 175-222, https://doi.org/10.5194/gmd-9-175-2016, 2016.

---

## Author Comment (AC2) · 27 Nov 2020

**Response to comments of reviewer 2**

Manuscript:      essd-2020-156
Original title:   Spatially-explicit estimates of global wastewater production, collection, treatment and re-use.
Revised title:   Country-level and gridded estimates of wastewater production, collected, treatment and re-use.
Authors:         Edward R. Jones, Michelle T.H. van Vliet, Manzoor Qadir, Marc F. P. Bierkens

**General comments from reviewer**

This study provided the comprehensive and consistent global outlook on the state of wastewater production collection treatment and reuse. And the country level wastewater data are downscaled and validated at 5 arc- minute resolution. Its results represent the first efforts to global wastewater collection treatment and reuse at the subnational level. It is a very interesting and useful work for the wastewater research. And the quality of the data set as submitted is high. The data analysis and discussions are sufficient. So I think it prepared well for publication. It analyzed the relationship among the production, collection, treatment and reuse of wastewater, the income level and the population.

We thank the reviewer for reading the manuscript and providing their feedback. We were very pleased to read that the reviewer thinks the dataset is of high quality and useful for wastewater research.

However, I think the influence from the pollution of agriculture, especially for the global grain production areas, cannot be ignored. So I suggest the author to add the analysis or discussion of this part. It may be more perfect.

We agree that agricultural runoff is a very important source of water pollution, and that we should reflect more upon this in the Discussion section of the manuscript. Whilst a more detailed analysis of diffuse and point source pollution from the agricultural sector would be beneficial for the water quality modelling community, this is outside the scope of this study, which solely focuses on municipal wastewater (i.e. domestic and industrial sources) for a number of reasons.

Firstly, country-level data is much more readily available for the municipal sector. This makes our (data-driven) approach more applicable for these sectors. Return flows from agricultural activities at the global scale are more typically quantified by modelling of irrigation water demand (net and gross) and withdrawal.

Secondly, agricultural runoff, which is an important source of pollution by the agricultural sector, is rarely collected or treated (e.g. WWAP, 2017) and hence far less applicable to this study. Conversely, collection (particularly sewers) and treatment infrastructure are very important for determining the fate of pollutants generated by municipal activities (e.g. collection and treatment infrastructure as point sources of pollutants, abatement of pollutant levels via treatment processes).

To address this comment, we propose to add the following lines to the discussion section of the manuscript:

*While agricultural runoff is also a substantial source of pollution, this is outside the scope of this study. Country-level data on agricultural runoff is sparse, necessitating modelling approaches to quantify irrigation return flow by calculating net demand (e.g. based on crop composition and irrigated area per grid cell), gross irrigation demand (to account for irrigation efficiency and losses) and water withdrawals (Sutanudjaja et al., 2018). Agricultural runoff is also rarely collected or treated (UNEP, 2016), hence is less applicable for inclusion in this study.*

**References**

Sutanudjaja, E., Beek, R., Wanders, N., Wada, Y., Bosmans, J., Drost, N., Ent, R., de Graaf, I., Hoch, J., de Jong, K., Karssenberg, D., López, P., Pessenteiner, S., Schmitz, O., Straatsma, M., Vannametee, E., Wisser, D., and Bierkens, M.: PCR-GLOBWB 2: A 5 arcmin global hydrological and water resources model, Geoscientific Model Development, 11, 2429-2453, 10.5194/gmd-11-2429-2018, 2018.

UNEP: A Snapshot of the World's Water Quality: Towards a global assessment, United Nations Environment Programme, Nairobi, Kenya, 162pp, 2016.

WWAP: The United Nations World Water Development Report 2017. Wastewater: The Untapped Resource, Paris, UNESCO, 2017.

---

## Author Response (AR1)

| Manuscript: | essd-2020-156 |
|---|---|
| Original title: | Spatially-explicit estimates of global wastewater production, collection, treatment and re-use |
| Revised title: | Country-level and gridded estimates of wastewater production, collection, treatment and re-use |
| Authors: | Edward R. Jones, Michelle T.H. van Vliet, Manzoor Qadir, Marc F. P. Bierkens |

**Response for the editor**

To the editor,

Thank you for consideration publication of this paper in ESSD. Please find our detailed response to the comments of the reviewers and the manuscript including tracked changes below. We also provide line numbers for the substantive text changes in the manuscript. Please note the line numbers refer to the version of the manuscript that includes the tracked changes.

Many thanks.

**Response to comments of reviewers**

**Reviewer #1**

**General comments from reviewer**

The paper by Jones et al. provides a revised and consistent global outlook of the state of wastewater production, collection, treatment and re-use. It uses available country-level wastewater data and regression analyses to estimate information where it is unavailable. The year selected for the country-level data was 2015, and unavailable data were standardized to the same year using relationships with GDP. In addition, the authors downscaled the country-level data to a 5-min resolution grid using return flow data from the global water balance model PCR-GLOBWB. The downscaling was validated using European, US, and some global (yet less in numbers) records of wastewater treatment plants. Validation efforts delivered reasonable model performance indicators, and uncertainties were estimated using a bootstrapping technique.

The final data product provides a gridded map that includes quantities of wastewater production, collection, treatment and re-use at 5-min spatial resolution. Global water quality models and large-scale water assessments have been lacking this type of information in the past, so this paper is clearly a very important addition to the field. As this is a global effort, there are severe constraints regarding data availability and quality. The authors developed (and explained) reasonable approaches to overcome these problems. While some of their methods are based on speculation regarding the relevant processes, the ultimate test of such an approach is the validation of the results. I think the authors did a commendable job in their validation and comparisons, and I do appreciate that they reveal important shortcomings and clearly state that the results must be interpreted with caution. So overall I think this paper presents an excellent global-scale effort to generate an advanced and novel gridded map of wastewater quantities. The manuscript is generally well written and very clearly structured. I strongly recommend its publication. I have a series of mostly minor comments that I list below. They are all written with the intention to further improve the manuscript. I also want to express my thanks to the authors for providing this important dataset to the research community!

We thank the reviewer very much for the insightful comments and suggestions. We are pleased to read that the reviewer is complimentary about the manuscript and recognises the value of our dataset for the research community. The reviewer addresses a number of important topics, which we overall agree with and which have all been addressed in the revised manuscript. Please see our point-by-point responses to the individual comments below.

**Individual comments**

1. Title (and elsewhere): After reading the title ("Spatially-explicit . . .") my initial expectation was that the paper will describe explicit locations of wastewater production and collection (e.g. locations of wastewater treatment plants). Only when reading the manuscript, I realized that the dataset refers to a modeled distribution of wastewater quantities at sub-national scale. This is still great, but maybe a slightly different title could help avoiding this confusion, such as "Downscaled gridded model estimates of global wastewater. . ." or you could at least refer to "model estimates" rather than just "estimates".

We understand this source of confusion for the reviewer. We have changed the provisional title to: "*Country-level and gridded estimates of wastewater production, collection, treatment and re-use*". We have chosen not to include the word 'modelled' in our dataset, as we do not want to give off the impression that the results are exclusively modelled (as the study is primarily underpinned by reported data at the country-level).

2. There is no distinction made in the dataset between industrial and domestic wastewater - which have very different characteristics and effects on environmental waters. It would thus be great to briefly discuss whether and how the combination of domestic and industrial wastewater may cause problems in the dataset, and in particular in the downscaling process. For example, industrial wastewater can be produced at locations with little correlation to population centers, i.e. at very distinct or remote locations compared to domestic wastewater - is this accounted for in the modeled return flows of PCR-GLOBWB? Also, I assume that the validation data cannot clearly distinguish between domestic and industrial wastewater as well? Related to this: Line 58 states for the first time that throughout the manuscript, domestic and industrial wastewater are not considered separately but lumped. As this is very important, it could be emphasized more, e.g. by referring more clearly to "combined domestic and industrial sources". This could also be done at other locations, where appropriate.

PCR-GLOBWB calculates water use (i.e. withdrawals, consumption and hence return flows) from the domestic and industrial sectors individually, which we have then lumped together for the downscaling procedure. Since the initial manuscript submission, we have further updated the return flows used for the downscaling procedure using a more recent water use dataset, developed at 5arc-min, from Water Futures and Solutions (WFaS) initiative (Wada et al., 2016). The use of this more recent dataset also facilitated an improved downscaling methodology and results.

In both the WFaS and PCR-GLOBWB methodologies, domestic demands are calculated on the basis of population and a country-specific per capita water use. Conversely, industrial demands are calculated on the basis of four socio-economic variables (GDP, electricity production, energy consumption and household consumption) (see Wada et al., 2011; Wada et al., 2014 and Wada et al., 2016 for details). Thus, industrial demand (and hence return flows) are simulated in areas taking into account multiple variables aside from just population. Regarding the question of the reviewer, the industrial return flows used do indeed account for industrial flows also being located in areas outside population centers.

We agree that the composition of industrial and domestic wastewater is different and that both have different environmental effects. We also strongly agree that a distinction between industrial and domestic wastewater is important for different applications. However, we choose to lump these flows for now. Domestic and industrial return flows are typically collected in the same (municipal) sewers before conveyance to wastewater treatment. As the reviewer points out, a distinction between domestic and industrial wastewater treatment can not be validated individually for this reason. We therefore prefer to present aggregated results of municipal wastewater, including both domestic and industrial wastewater.

The manuscript has been updated as appropriate to reflect these changes in using this more recent WFaS water use dataset (Wada et al, 2016).

Lines 207- 210: *"Return flows used for downscaling are calculated as gross - net water demands from the Water Futures and Solutions (WFaS) initiative for the years 2000 – 2010 (Wada et al., 2016). The WFaS water demand dataset follows the approach developed for PCR-GLOBWB (Wada et al., 2014)."*

Lines 217 – 226: *"Wastewater collection is assigned sequentially to grid cells with the largest downscaled produced wastewater flows. Thus, collected wastewater is preferentially allocated to grid cells with the highest levels of municipal activities, where central wastewater collection (and treatment) is assumed to be most economically feasible. Wastewater treatment is assigned to grid cells only where wastewater collection exists, at an average treatment rate calculated at the country-level. The treatment rate is calculated as the proportion of collected wastewater that undergoes treatment, and hence can differ from the country-level wastewater treatment percentage (which is calculated as the proportion of produced wastewater that is treated). For the downscaling of wastewater re-use an additional criterion was introduced to represent water scarcity, a key driver of wastewater re-use. The ratio of water demand to water availability was calculated. Grid cells within a country with a treated wastewater allocation are then ordered based off this ratio and treated wastewater re-use was assigned sequentially to these grid cells."*

3. Lines 62 and following: You state that "wastewater treatment improves the quality of 'used' water resources" and this notion seems to prevail throughout the introduction and discussion. But while "wastewater treatment" as a process certainly has the GOAL to improve water quality, what about the fact that substances that are not or cannot be treated by treatment facilities can cause the opposite effect: in these cases, wastewater treatment plants can represent point sources of pollution, especially in the case of emerging contaminants. This has not been addressed in the paper and I thus encourage the authors to at least briefly reflect on the issue.

This is an excellent point and we entirely agree that this should be considered in the manuscript. We recognise that wastewater collection and treatment processes, if insufficient, can concentrate particular pollutants (especially for emerging pollutants) and thus represent a point source for environmental contamination. We have added some sentences in the discussion section to reflect this:

Lines 494 – 503: *"It should be noted that while the aim of wastewater collection and treatment is to reduce pollutant loadings to minimise risks to human health and the environment, these facilities can also act as point sources of pollution. Wastewater collection concentrates pollutants which, can pose serious water quality issues if discharged with insufficient treatment. Furthermore, a range of emerging pollutants (e.g. pharmaceuticals, pesticides and industrial chemicals) are concentrated in wastewater collection networks (Geissen et al., 2015). These pollutants are of particular concern as they are not typically monitored for or sufficiently removed in wastewater treatment processes, with ambiguous risks posed to human and environmental health even in low concentrations (Deblonde et al., 2011; Geissen et al., 2015). The solution is not however to collect less wastewater, but to increase*

*treatment in terms of percentage of collected wastewater, treatment level and the number of pollutants (UNEP, 2016)."*

4. Table 1 (Standardisation to 2015): There is very little explanation in the text about the rationale behind the standardization methods. It seems the main assumption is a linear behavior of wastewater amounts based on GDP, right? This could briefly be mentioned in the text. Also, for collection and treatment: I do not understand why the values are divided by GDP per capita but then not multiplied by GDP per capita but by total GDP. Is this just a typo?

For the standardisation to 2015, we indeed assumed a linear behavior with on GDP (for wastewater production) and GDP per capita (for wastewater collection and treatment). We have added a line to clarify and justify this choice in the text:

Lines 140 – 144: "*Wastewater production is assumed to be dependent upon both population size and per capita production (related to per capita wealth). Hence, we standardise wastewater production linearly with GDP, a combined metric of population size and wealth. Conversely, wastewater collection and treatment are assumed to be more dependent on economics, hence we linearly apply GDP per capita for standardisation*".

We also would like to thank the reviewer for bringing the issue with GDP vs. GDP per capita to our attention. This is indeed an unfortunate typo. We have made the corrections to Table 1 (at line 155) and confirm that GDP per capita is indeed the correct variable that was used for standarising collection and treatment.

5. Figure 3: Very interesting figure. The one country that stands out to me as a surprise in wastewater production is Egypt. The Nile and Nile delta show also exceptionally high values in Figures 4 and 6. There is no comment about Egypt in the manuscript. Any explanations on why Egypt has so high domestic and industrial wastewater amounts?

We agree that Egypt is a particularly interesting country, and stands out in Figures 4 and 6 (which are updated, with the WFaS 5-arcmin water use dataset, but shows the same pattern.

It is worth noting that wastewater data for Egypt was cross referenced from three sources: 1) GWI 2015: 13,623 million $m^3$ $yr^{-1}$; 2) UNSD 2015: 11,899 million $m^3$ $yr^{-1}$; and 3) Aquastat 2012: 6,497 million $m^3$ $yr^{-1}$ which, standarised to 2015, was 8,429 million $m^3$ $yr^{-1}$. Whilst some variation exists between these numbers, data from all three sources indicate a relatively high per capita wastewater production (between 91 – 147 $m^3$ $yr^{-1}$ per capita). The final value determined (mean average from the three sources for 2015 data) for Egypt as 11,317 million $m^3$ $yr^{-1}$ (122 $m^3$ $yr^{-1}$ per capita) – which is relatively high (compared, for instance, to regional averages). This is further validated by estimates of Egypt's domestic water use (200 l per capita per day in 2007), almost double that of Germanys (data from National Water Research Centre (Egypt).

In terms of the downscaled maps (Figures 4 and 6), population density is an important variable for quantification of domestic and industrial return flows in PCR-GLOBWB. Egypt's population is of course heavily concentrated along the banks of the Nile and in the Nile Delta. This heavily concentrates the downscaled wastewater into relatively few gridcells (relative to the size of Egypt). Next to this, we expect that the domestic + industrial return flows (or wastewater) values may also be high due to the relatively low water use efficiencies and system losses.

6. To my knowledge, the European dataset of wastewater treatment plants reports treatment capacity only in Population Equivalent (PE); however, the manuscript states that the volume flow rate was obtained based on a linear regression for plants reporting both parameters (PE and volume). It would be interesting to know how many plants included this information since the openly available dataset at the EEA website seems not to provide the volume flow rate.

The reviewer makes a good point here with regards to our need to convert wastewater treatment in population equivalent to volume flow rate for validation purposes. We indeed used a linear relation between Population Equivalent (PE) and volume flow rate for wastewater treatment plants that reported both of these variables, and applied this to plants where only the PE was reported.

Fortunately, we were able to obtain a decent number of data points both for the global database (GWI) and specifically for Europe (EEA) in order to do the linear regression.

For the EEA dataset, both PE and volume flow rate (in $m^3$ $yr^{-1}$) data were available for 8,309 treatment plants. The linear relationship was applied to 17,593 plants with only treatment capacity in PE, to estimate treatment capacity in volume flow rate units. Predicted vs. reported wastewater treatment capacity is displayed for the EEA plants where both PE and volume flow rate units (e.g. $m^3$ $yr^{-1}$) are reported in Figure R1 ($R^2 = 0.80$).

For the GWI dataset, this was available for 227 wastewater treatment plants, with the linear relation applied to 83 plants with capacity only reported in PE. The remaining plants has their capacities only reported in volume flow units. It is important to note that wastewater treatment plants located in the US and Europe were excluded from the GWI dataset to avoid plants potentially being counted multiple times if contained in the GWI dataset + the EEA or US EPA dataset. Predicted vs. reported wastewater treatment capacity is displayed for the GWI plants where both PE and volume flow rate units (e.g. $m^3$ $yr^{-1}$) are reported in Figure R2 ($R^2 = 0.81$).

All wastewater data from the US EPA were reported in volume flow rate units (million gallons per day) which was converted into $m^3$ $yr^{-1}$.

With regards to the EEA data, we found this data to be freely available. If the reviewer is interested in this dataset, they can download this information from Waterbase (https://www.eea.europa.eu/data-and-maps/data/waterbase-uwwtd-urban-waste-water-treatment-directive-6) and consult the worksheet named "UWWTPS". Providing this dataset has not since been adjusted, information for plant latitude (column AF); longitude (column AH); load entering UWWTP in PE (column AG) and wastewater treated (column BP) in $m^3$ $yr^{-1}$ can be obtained. Alternatively, the reviewer is welcome to contact us directly if they would like the plant level data for the EU. As part of this paper, we have already collated important information for them various spreadsheets provided by the EEA into a single spreadsheet and undertaking some basic formatting.

[Figure]

**Figure R1.** Predicted vs. reported wastewater treatment for EU wastewater treatment plants reporting treatment capacity in both population Equivalent (PE) and in volume flow rate units (e.g. million m$^3$ yr$^{-1}$ ).

[Figure]

**Figure R2.** Predicted vs. reported wastewater treatment for GWI wastewater treatment plants reporting treatment capacity in both population Equivalent (PE) and in volume flow rate units (e.g. million $m^3$ $yr^{-1}$ ).

7. Line 437 and following: You say: "This may occur due to discrepancies between the design (i.e. maximum) capacity of wastewater treatment plants . . ." Ok, that is a fair point. But the other option is that your model overpredicts in places without treatment plants and therefore underpredicts in places with treatment plants. This potential model bias should be acknowledged and briefly discussed.

We agree that this potential bias might also explain these discrepancies. We think this is ultimately depends on the downscaling method for wastewater production (i.e. using modelled return flow data from PCR-GLOBWB), on which quantifications of wastewater treatment are based. We have tried to explain this in more detail in the text:

Lines 437 – 441: *"Furthermore, uncertainties in the data used as basis for downscaling of wastewater production (i.e. PCR-GLOBWB return flows) directly impacts the downscaled results of wastewater treatment. For example, the underprediction of return flows in urban areas and overprediction in rural areas could lead to the overprediction of wastewater treatment in areas without treatment plants and underprediction of wastewater treatment for grid cells with large treatment capacities."*

8. Line 244-245: "For validating downscaled wastewater re-use, only plants (with treatment capacity > 1 million m3 yr-1) using tertiary or higher wastewater treatment technologies were considered." The rationale for this decision is not clear to me.

Unfortunately, plant-level data on wastewater re-use is limited. Due to the lack of data, assumptions had to be made for validating wastewater re-use. Aside for the limited number of plants specifically

designed as re-use facilities from GWI (78 wastewater re-use plants), we assumed that tertiary or higher wastewater treatment processes likely are most strongly related to wastewater re-use (as these treatment levels are a pre-requisite for safe re-use for most applications). We narrowed the scope to larger wastewater treatment plants only (>1 million m$^3$ yr$^{-1}$), where tertiary+ treated wastewater is potentially available for re-use in substantial volumes and to exclude highly localised wastewater treatment facilities (e.g. for specific industrial facilities) which are very difficult to capture in lieu of poor data availability.

9. Figure 1 shows that there are data for only ~half of global countries available (e.g. production data are available for 118 countries), yet 90% of population is covered by data. The only explanation for this is that all the high-population countries are included in those countries for which data exist, correct? It may be worth pointing this out as initially I was confused on how these numbers match up. I found some explanations later in the manuscript, but maybe this general fact could be stated earlier.

The reviewer is correct. Reported data was typically more available for large countries as opposed to smaller ones. We have tried to make this more clear in the text when the comparison between % population and number of countries with reported data is first mentioned.

Lines 147 – 148: *"Reported wastewater data was available for the majority of the world's most populous countries. This results in a the high percentage population coverage relative to the number of countries."*

**Minor Comments**

Throughout the text, the authors use the expression "data" in singular form ("data is . . ."). I am more used to data in plural form ("data are . . .").

We agree with the reviewer. This has been corrected throughout the manuscript.

Also throughout the text, the authors use the expression "whilst" (many times). Its my understanding that "whilst" may be perceived as 'archaic' in American English (https://en.wikipedia.org/wiki/While). So maybe use "while" instead?

We have changed all occurrences of "whilst" to "while", the more familiar formulation of the word in American English.

Line 24 (and possibly elsewhere): The expression "significant" is often reserved for instances where it refers to statistical methods. Here, an alternative might be to use "substantial".

We thank the reviewer for this comment, and agree that "substantial" is a better word choice for instances not referring to statistical methods. This has been adjusted throughout the manuscript.

Line 25: replace "containing" with "comprising"?

This has been changed.

Line 28: I suggest spelling out the first occurrence of SDG here (or remove the example)

We have removed the example of the SDGs in the abstract for readability. The anacronym SDG is now spelt out in the first occurrence in the manuscript instead.

Line 80: "that ensure" instead of "to ensure"?

This has been changed.

Line 88: rephrase "whereby . . ." - maybe "which includes a target that . . ."

This line has been reformulated in line with the recommendation to rephrase.

Line 89: add a space in "SDG 6.3"

This has been changed.

Line 98: say "to a grid level of 5 arc-minute spatial resolution"

This has been changed.

Figure 1: I cannot find the letters (a), (b) and (c) reflected in the figure, so it is difficult to find out which panels the caption refers to. Also, I suggest removing the asterisk from the figure and simply add the definition of 'population coverage' as part of the normal caption.

Thanks for noticing that (a), (b) and (c) was missing from the figure. This was a mistake and has now been corrected. The asterisk has been removed as recommended, and the figure caption has been updated. Figure panels b and c have also been increased slightly in size for better readability.

Line 117: add "(Table 1)" after ". . . databases"

This has been added.

Lines 125-126 (and elsewhere): I find the use the acronyms, the article "the", as well as the verbs not consistent here. I would say "GWI reports . . . whereas FAO reports . . . and UNSD reports" etc.

This has been changed in the suggested way.

Line 126: say ". . . reported by UNSD"

This should read more clearly now with the changes to lines 125-126.

Lines 139-140: repetitive use of "both" - delete one?

This has been changed.

Table 1: start title with "Wastewater data sources and population coverage by region and economic aspects. . ." Also, the footnotes of the table could be shortened. E.g. the square brackets always refer to the number of countries, so this could be explained once in the title and does not need to be repeated for each of the footnotes.

Thanks for this suggestion to alter the title and footnotes. These have been changed.

Line 166 (and possibly elsewhere): there are some instances where "per capita" is written as "per Capita" (capitalized)

"per capita" (lowercase) is now consistently used throughout the manuscript.

Line 168-169: "Data was transformed, as appropriate, to ensure normality." This statement is not clear to me. Does it refer to what is called "sqrt" in Table 3, which I guess means that the square-root of values was calculated? Could both instances be clarified, e.g. by adding a little more information in this sentence here?

Some variables (e.g. GDP, population) were log-transformed to limit skew in the independent variables (which can vary across many orders of magnitude) and to achieve normality. Alternatively, square-root (sqrt) transformation is used for desalination capacity per capita as this data-set contained zero values (countries with no desalination) and thus is inappropriate for log transformation. We have further clarified this in the text:

*Lines 175 – 176: "Data was transformed, either using a log or square root transformation, to reduce the skew in the independent variables and to ensure normality."*

Table 2: spelling of "Agricultural Land" should be "Agricultural land"

Thanks for noticing this – the appropriate change has been made.

Line 201: The eight regions are listed here for the first time, but no explanation is provided on what exactly defines these regions. Is this some official breakdown so that one could look up the countries that belong to each region?

Thanks for this. The regional classifications are taken from the World Bank, the same as for the countries and income classifications. We agree this should be more clear in the text, so we have added this.

Line 215: "occur" instead of "occurs"

This has been changed.

Line 241: repetition of "both"

One occurrence of the word "both" has been removed.

Line 243: "where aggregated PER CELL"?

Correct. This has been added.

Line 248: "were" instead of "was"

This has been changed.

Line 254 and 257: While most explanations in this paragraph are in the past tense, two verbs are in present tense ("is expanded" and "is then compared"). Typo?

Thanks for noticing this. This has been changed.

Line 261: "were" instead of "was"

This has been changed.

Line 289: "Human Development Index" (as it was also capitalized elsewhere)

This has been changed.

Line 290-291: say "was found to have the strongest influence on"

Good suggestion. This has been changed.

Line 311: "nations" (plural)

This has been changed.

Figure 2: Could add the number of countries that are displayed in each panel, e.g. add "n = ...". Also, the graphs may be clearer if using white instead of grey background.

We have added the number of countries in each panel, as suggested. We prefer to keep the backgrounds of the graphs grey.

Line 326: move "billion" after parentheses

This has been changed.

Line 331: "being from" instead of "from"

This has been changed.

Line 353: "World Health Organization's" (with apostrophe)

This has been changed.

Line 362: "indicate" instead of "indicates"

This has been changed.

Line 379: add commas before and after "respectively"

This has been changed.

Line 391: delete "regions"

This has been changed.

Line 397: "treat" instead of "treated"

This has been changed.

Line 398: "than in" instead of "than by"

This has been changed.

Table 4, title: Start with "Wastewater production, collection, treatment and re-use (billion m3 yr-1) by region and economic development level." I suggest changing "brackets" to "parentheses" (as brackets in American English would refer to squared brackets). And say "regressions" (plural).

Thanks for this suggestion. We have made altered the title as recommended.

Line 421: "yr-1" instead of "/year". Also, "occur" instead of "occurs"

This has been changed.

Line 422: "collection . . . and treatment . . . are" (instead of "is")

This has been changed.

Line 426: "with only available wastewater resources" not clear, wrong wording?

We agree this is ambiguous wording. We have made the appropriate changes in the text to clarify this.

Line 456: I suggest using "underpinning source data" instead of "underlying source data"

This has been changed.

Line 472: "upon which country-level estimates incorporate" seems to be incorrect wording

We agree this is incorrectly worded. We have changed the wording in the manuscript.

Line 479: I suggest breaking this very long paragraph into 2 here

This has been changed.

Line 486: say "factors" instead of "drivers" to avoid repetition

This has been changed.

Line 489: say "of untreated" (add "of")

This has been changed.

Line 496: "Whilst our results also rely on this approach, we instead used. . ." This sounds odd (first you say you do the same, then you say 'instead') - rephrase?

We agreed that this is an 'odd' statement. We have adjusted the wording to make this more clear.

Line 512: The expression "acreage" may not be known to all readers. Could just say "spatial extent"

We prefer to keep the term "acreage" here, as this is the terminology used in the referenced work.

Line 519: close the parentheses

This has been changed.

Line 522: "for as a baseline for"? should this be "as a baseline for"?

This has been changed.

Line 525: ". . . problems of discrepancies in data reporting years and missing data are overcome." It sounds quite optimistic that the problems are truly "overcome" (i.e. solved). Maybe say "reduced" instead?

Good suggestion. We now use the word "reduced" instead.

Lines 528-529: repetitive use of "particularly"

This has been changed.

Lines 549-551: you use the word "such" three times here

This has been changed.

Lines 550-551: The grammar/verb of the description of point (4) seems incorrect. Change "creating" to "create"?

This has been changed.

**References: Reviewer 1**

Deblonde, T., Cossu-Leguille, C., and Hartemann, P.: Emerging pollutants in wastewater: a review of the literature, International journal of hygiene and environmental health, 214, 442-448, 10.1016/j.ijheh.2011.08.002, 2011.

Geissen, V., Mol, H., Klumpp, E., Umlauf, G., Nadal, M., van der Ploeg, M., van de Zee, S. E. A. T. M., and Ritsema, C. J.: Emerging pollutants in the environment: A challenge for water resource management, International Soil and Water Conservation Research, 3, 57-65, https://doi.org/10.1016/j.iswcr.2015.03.002, 2015.

Wada, Y., Beek, L. P. H., Viviroli, D., Dürr, H., Weingartner, R., and Bierkens, M. F. P.: Global monthly water stress: II. Water demand and severity of water, Water Resources Research - WATER RESOUR RES, 47, 10.1029/2010WR009792, 2011.

Wada, Y., Wisser, D., and Bierkens, M. F. P.: Global modeling of withdrawal, allocation and consumptive use of surface water and groundwater resources, Earth Syst. Dynam., 5, 15-40, https://doi.org/10.5194/esd-5-15-2014, 2014.

Wada, Y., Flörke, M., Hanasaki, N., Eisner, S., Fischer, G., Tramberend, S., Satoh, Y., van Vliet, M. T. H., Yillia, P., Ringler, C., Burek, P., and Wiberg, D.: Modeling global water use for the 21st century: the Water Futures and Solutions (WFaS) initiative and its approaches, Geosci. Model Dev., 9, 175-222, https://doi.org/10.5194/gmd-9-175-2016, 2016.

**Reviewer #2**

**General comments from reviewer**

This study provided the comprehensive and consistent global outlook on the state of wastewater production collection treatment and reuse. And the country level wastewater data are downscaled and validated at 5 arc- minute resolution. Its results represent the first efforts to global wastewater collection treatment and reuse at the subnational level. It is a very interesting and useful work for the wastewater research. And the quality of the data set as submitted is high. The data analysis and discussions are sufficient. So I think it prepared well for publication. It analyzed the relationship among the production, collection, treatment and reuse of wastewater, the income level and the population.

We thank the reviewer for reading the manuscript and providing their feedback. We were very pleased to read that the reviewer thinks the dataset is of high quality and useful for wastewater research.

However, I think the influence from the pollution of agriculture, especially for the global grain production areas, cannot be ignored. So I suggest the author to add the analysis or discussion of this part. It may be more perfect.

We agree that agricultural runoff is a very important source of water pollution, and that we should reflect more upon this in the Discussion section of the manuscript. Whilst a more detailed analysis of diffuse and point source pollution from the agricultural sector would be beneficial for the water quality modelling community, this is outside the scope of this study, which solely focuses on municipal wastewater (i.e. domestic and industrial sources) for a number of reasons.

Firstly, country-level data is much more readily available for the municipal sector. This makes our (data-driven) approach more applicable for these sectors. Return flows from agricultural activities at the global scale are more typically quantified by modelling of irrigation water demand (net and gross) and withdrawal.

Secondly, agricultural runoff, which is an important source of pollution by the agricultural sector, is rarely collected or treated (e.g. WWAP, 2017) and hence far less applicable to this study. Conversely, collection (particularly sewers) and treatment infrastructure are very important for determining the fate of pollutants generated by municipal activities (e.g. collection and treatment infrastructure as point sources of pollutants, abatement of pollutant levels via treatment processes).

To address this comment, we have added the following lines to the discussion section of the manuscript:

[revised manuscript text omitted]